# Cluster hadronisation in SHERPA

**Gurpreet Singh Chahal and Frank Krauss**

Institute for Particle Physics Phenomenology, Durham University, Durham DH1 3LE, UK

## Abstract

We present the SHERPA cluster hadronisation model and a simple model for non-perturbative colour reconnections. Using two different parton shower implementations we tuned the model to data and we show typical resulting distributions that are sensitive to hadronisation effects in $e^+e^-$–annihilations into hadrons.

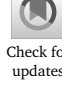 Check for updates

# 1 Introduction

The confinement property of the strong interaction implies that only hadrons are directly observable at the low scales accessible in detectors. To arrive at a detailed description of particle reactions at collider experiments such as the LHC necessitates the use of models for the transition from the fundamental QCD particles, quarks and gluons, to their bound states, the hadrons. This transition, known as hadronisation, is responsible for the bulk of particle production in events involving the strong interaction, and it has directly observable consequences on quantities such as energy flows, jet shapes, or rapidity gaps. In the absence of a quantitative understanding based on the first principles, hadronisation is being described in terms of phenomenological models and usually embedded within event generators [1].

In the perturbative part of the simulation, the event generators describe the production of quarks and gluons, partons, in a sequence of stages; starting with exact, fixed-order matrix elements at the largest momentum scales, these primary hard partons are successively dressed through the emission of softer or collinear secondary partons at decreasing scales in the parton showers. In hadron collisions, multiple parton-parton interactions in the underlying event further increase the number of partons, an effect again described through phenomenological models. Ultimately, the perturbative part of the event generation results in parton configurations resolved at scales of the order of a GeV. At this point, hadronisation models take over and turn the partons into sets of primordial hadrons, some of which may be unstable and must decay further.

Broadly speaking, currently used hadronisation models fall into two categories. Building on linear confinement and the idea of practically one-dimensional QCD flux tubes, string models were first discussed in [2]; a powerful realisation of the string idea, and the most famous one, known as the Lund model, has been worked out in [3,4]. It has been implemented and subsequently further refined in the PYTHIA event generator [5,6]. In the past decade, the

model has been extended to the notion of colour "ropes" [7–10], fused strings, which are of particular importance in heavy-ion collisions and better account for effects such as strangeness enhancement or collective flows. More recently, a thermodynamic approach to string fragmentation has been studied in [11], modifying, among others, production rates of heavy hadrons. Time-dependent string tension was shown to increase the rates and modify the kinematics of strange and baryon production [12], and similarly, hyper-fine splittings in hadron formation in the string model have been analysed in [13], which also affect the production yields of hadron species, in particular strange hadrons.

In contrast, local parton-hadron duality (LPHD) [14] and, in particular, preconfinement [15] have been the guiding principles underpinning the development of cluster hadronisation models in [16–19] and in [20, 21]. The latter has been implemented in the HERWIG event generator [22, 23].

In the following the hadronisation model of SHERPA [24–26] will be presented. It builds on the original independent realisation of the cluster model idea in [27] and has been further refined in a new and improved implementation. We will describe its underlying principles in detail in Section 2 and we will highlight some of the considerations in resolving problematic kinematic configurations, typically involving extremely light clusters. In Section 3 we will introduce a first, simplistic model for colour reconnections in SHERPA. It provides an alternative to models motivated by the analysis of such effects in the measurements of the $W$ mass [28] and the top mass [29], following on a first implementation in the framework of the description of multiple–parton scattering in [30]. By and large, models such as the one in [30] and its extensions or variations in PYTHIA [31, 32] and HERWIG [33–37], correct for the effect of interpreting the colour flow in parton showers through planar diagrams [38], *i.e.* the unique, one–to–one relation of colours and anti-colours, and they also include collective effects, for example through the colour ropes mentioned above.

We turn to the presentation of results in Section 4, obtained by tuning the model with and without colour reconnections for two parton showers, CSShower [39] and DIRE [40] implemented in SHERPA. The details on the tuning parameters are given in Appendix B. The performance of new tunes at different energies is discussed in Section 5. We conclude in Section 6 with an outlook.

## 2 Cluster hadronisation Model

### 2.1 Cluster formation

After the parton shower evolution stops at transverse momenta $p_{T,\text{min}} \approx 1$ GeV, hadronisation models take over and transform the resulting partons, quarks and gluons, into primary hadrons, some of which may decay further. In cluster fragmentation models this is achieved by forming colourless clusters made of quarks and anti-quarks. This results in a forced, non-perturbative splitting of each of the gluons into a quark–anti-quark pair, carrying its two colours. Since typical parton showers are formulated in the limit of infinitely many colours, $N_c \to \infty$, each coloured quark can thus be associated with an anti-quark of the exact anti-colour; the colour–anti-colour pair then neutralises each other by forming a colour-singlet cluster. In SHERPA, as in HERWIG and PYTHIA [41], baryons are assumed to be composed of a quark $q_1$ and a diquark $(q_2 q_3)$, the (fictitious) bound state of two quarks, such that any baryon $B = q_1(q_2 q_3)$. In SHERPA these diquarks can already emerge in the gluon splitting, *i.e.* during the formation of the primary clusters. This results in a somewhat softened correlation of $B\bar{B}$ pairs in phase space, similar to the popcorn mechanism in PYTHIA [42, 43]. To ease the language we will collectively denote quarks and anti–diquarks emerging in the non-perturbative

phase of hadronisation as "flavours", with the implicit understanding that they will have a non-vanishing constituent mass, in contrast to the current quarks in the perturbative phases of event generation, such as the parton shower.

### 2.1.1 Non-perturbative splitting of gluons

Quark and gluon ensembles produced in the large-$N_c$ limit by parton shower are colour-ordered sequences of a flavour (quark or anti–diquark), some gluons, and an anti-flavour (anti–quark or diquark), or of gluons only. Postponing a discussion of the latter case to a later stage, let us first focus on the non-perturbative splitting of the gluons in such an $n$-particle sequence $f_1 g_2 g_3 g_4 \ldots g_{n-1} \bar{f}_n$.

### 2.1.2 Splitting a gluon

In SHERPA the gluons in these sequences are split step-wise from the edges, $g \to \bar{\tilde{f}} \tilde{f}$. The resulting anti-flavour or flavour combines with the neighbouring flavour or anti-flavour into a cluster $\mathcal{C}[f \bar{\tilde{f}}]$ or $\mathcal{C}[\tilde{f} \bar{f}]$, schematically,

$$
f_1 g_2 g_3 g_4 \cdots \times
$$
$$
\times\, g_{(n-2)} g_{(n-1)} \bar{f}_n \to
\begin{cases}
\mathcal{C}[f_1 \bar{\tilde{f}}_{2'}] + \tilde{f}_{2''} g_3 g_4 \cdots g_{(n-2)} g_{(n-1)} \bar{f}_n & \text{for} \quad g_2 \to \bar{\tilde{f}}_{2'} \tilde{f}_{2''} \\
f_1 g_2 g_3 g_4 \cdots g_{(n-2)} \bar{\tilde{f}}_{(n-1)'} + \mathcal{C}[\tilde{f}_{(n-1)''} \bar{f}_n] & \text{for} \quad g_2 \to \bar{\tilde{f}}_{(n-1)'} \tilde{f}_{(n-1)''}
\end{cases}
,
$$

$$(1)$$

depending on whether gluon 2 or gluon $(n-1)$ was selected to split. This selection is taken at random unless either $f_1$ or $\bar{f}_n$ are heavy flavours, $c$ or $b$ quarks: in this case the "neighbour" gluon is selected (*i.e.* $g_2$ if $f_1$ is a heavy quark, and $g_{(n-1)}$ if $\bar{f}_n$ is a heavy quark). In the following we will assume that $g_2$ is the splitting gluon, the "splitter", and we denote flavour $f_1$ as "spectator". SHERPA then produces trial splittings of the gluon - determined by the selection of the produced flavour $\tilde{f}$ and the corresponding kinematics until an allowed solution is found.

### 2.1.3 Determining the flavour $\tilde{f}$ in gluon splitting

The produced trial flavour $\tilde{f}$ is selected according to the "popping" probabilities $P_{\tilde{f}}$, with available flavours subject to the constraint

$$
M_{12} - m_{f_1} \geq 2 m_{\tilde{f}} \,,
$$

$$(2)$$

where $M_{12} = \sqrt{(p_1 + p_2)^2}$ is the invariant mass of the splitter–spectator system. The $P_{\tilde{f}}$ are calculated from the input parameters according to Eq. (48), and include also the possibility of gluons decaying directly into diquarks.

### 2.1.4 Fixing the kinematics of the decay

In the rest frame of the splitter–spectator system, the splitter 2 is oriented along the negative $z$ axis and the spectator 1 is oriented along the positive $z$-axis,

$$
p_1^\mu = \frac{M_{12}}{\sqrt{2}} \left[ z_+ n_+^\mu + (1 - z_-) n_-^\mu \right] \quad \text{and} \quad p_2^\mu = \frac{M_{12}}{\sqrt{2}} \left[ (1 - z_+) n_+^\mu + z_- n_-^\mu \right],
$$

$$(3)$$

where

$$
z_+ = 1 \quad \text{and} \quad z_- = 1 - \frac{m_{f_1}^2}{2 M_{12}^2} \,,
$$

$$(4)$$

and the two light-like vectors $n^\mu_\pm = (1, 0, 0, \pm1)$.

The four-momenta of the spectator 1 and the two produced flavours 2' and 2" emerge from $p_1 + p_2 \to \tilde{p}_1 + \tilde{p}_{2'} + \tilde{p}_{2''}$ after the gluon splitting. Demanding that the spectator keeps its direction in the rest frame of the system, *i.e.* that its four momentum is entirely spanned by $n_\pm$ and demanding that the transverse momentum is compensated between the two new flavours yields a parameterization of the decay through

$$
\begin{aligned}
\tilde{p}^\mu_1 &= M_{12}\Big[ \qquad\quad x z^{(0)} n^\mu_+ \;+\; (1-z^{(1)})(1-y)\, n^\mu_- \Big], \\
\tilde{p}^\mu_{2'} &= M_{12}\Big[ (1-x)z^{(0)} n^\mu_+ \;+\; \qquad\quad (1-z^{(1)})y\, n^\mu_- \Big] \;+\; k^\mu_\perp, \\
\tilde{p}^\mu_{2''} &= M_{12}\Big[ (1-z^{(0)})\, n^\mu_+ \;+\; \qquad\qquad z^{(1)}\, n^\mu_- \Big] \;-\; k^\mu_\perp.
\end{aligned}
\tag{5}
$$

The absolute value $k_T$ of the transverse momentum $k_\perp$ is selected according to a Gaussian, with a maximum value given by the parton-shower cut-off, $p_{T,\mathrm{min}}$,

$$
\mathcal{P}(k_T) = \exp\Big[ -k_T^2/k_{\perp,0}^2 \Big] \Theta(p_{T,\mathrm{min}}^2 - k_T^2),
\tag{6}
$$

and its azimuth is flat, $k^\mu_\perp = k_T(0, \cos\phi, \sin\phi, 0)$. This leaves the determination of the longitudinal momenta fractions. The parameter governing the splitting of the gluon is $z^{(1)}$ and for its determination SHERPA offers two parameterizations, namely

$$
\mathcal{P}(z) =
\begin{cases}
z^\alpha + (1-z)^\alpha & \text{additive} \\
z^\alpha (1-z)^\alpha & \text{multiplicative}
\end{cases}.
\tag{7}
$$

From $k_T^2$ and $z^{(1)}$ the other kinematic parameters $z^{(0)}$, $x$, and $y$ are determined as

$$
\begin{aligned}
z^{(0)} &= 1 - \frac{m_{\tilde{f}}^2 + k_T^2}{z^{(1)} M_{12}^2}, \\
x &= \frac{Q^2 + m_{f_1}^2 - k_T^2 + \sqrt{(M^2 - m_{f_1}^2 - k_T^2)^2 - 4 m_{f_1}^2 k_T^2}}{2Q^2}, \\
y &= \frac{k_T^2}{(1-x)Q^2},
\end{aligned}
\tag{8}
$$

where $Q^2 = z^{(0)}(1-z^{(1)})M_{12}^2$. $z^{(0)}$ and both $x$ and $y$ have to be between 0 and 1 for the gluon splitting to be kinematically viable and, therefore, accepted. Once the kinematics has been fixed, particles 1 and 2' combine into a cluster $\mathcal{C}$, and 2" becomes a new spectator for the next splitting.

### 2.1.5 Gluon "rings"

In some cases, for example in the decay of heavy quarkonia such as $\eta_c \to gg$ or $J/\psi \to ggg$ or quite often in hadron collisions, the colour-singlet structures emerging from the parton shower and entering hadronisation are purely gluonic, $g_1 g_2 \ldots g_{(n-1)} g_n$. In this case SHERPA selects the colour-connected gluon pair with the largest combined invariant mass and splits one of the two gluons, $g_i$, with the other gluon acting as spectator. The resulting structure is then re-ordered to the form $f_i g_{(i+1)} g_{(i+2)} \cdots g_n g_1 g_2 \cdots g_{(i-1)} \bar{f}_{i'}$.

### 2.1.6 Clusters directly transiting to hadrons

Some of the primary clusters produced in the non-perturbative gluon splittings have masses $M_{\mathcal{C}}$ below the threshold $M_{\mathrm{trans}}[f_1 \bar{f}_2]$ for their direct transition to hadrons with the same flavour

quantum numbers. This threshold is given by a linear combination of the lightest and heaviest hadron mass as

$$M_{\text{trans}}[f_1\bar{f}_2] = x_{\text{trans}}\min(m_{\mathcal{H}[f_1\bar{f}_2]}) + (1 - x_{\text{trans}})\max(m_{\mathcal{H}[f_1\bar{f}_2]}), \tag{9}$$

with a tuning parameter $x_{\text{trans}}$. If $M_{\mathcal{C}} < M_{\text{trans}}[f_1\bar{f}_2]$ then the gluon splitting will directly produce a hadron instead of a cluster, with the hadron selected according to relative probabilities given by

$$\mathcal{P}_{\mathcal{C}[f_1\bar{f}_2] \to \mathcal{H}_1[f_1\bar{f}_2]+\gamma} = w_{\mathcal{H}} \left| \psi_{\mathcal{H}}(f_1\bar{f}_2) \right|^2 . \tag{10}$$

Here, $w_{\mathcal{H}}$ is a relative probability for the production of a hadron $\mathcal{H}$ – typically a combination of an overall multiplet weight and a hadron-specific additional modifier for certain "tricky" hadrons such as $\eta$ and $\eta'$ mesons – and $\psi_{\mathcal{H}}(f_1\bar{f}_2)$ is the flavour wave function of the hadron. In this case the gluon splitting kinematics of Eq. (5) is replaced with

$$\begin{aligned}
\tilde{p}_h^\mu &= M_{12}\Big[ & z^{(1)} n_+^\mu &+ (1-z^{(2)})n_-^\mu \Big] &+ k_\perp^\mu, \\
\tilde{p}_{2''}^\mu &= M_{12}\Big[ (1-z^{(1)}) n_+^\mu &+ & z^{(2)}n_-^\mu \Big] &- k_\perp^\mu,
\end{aligned} \tag{11}$$

with the updated values for $z^{(1,2)}$ given by

$$z^{(1)} = \frac{M_{12}^2 + m_{\mathcal{H}}^2 - m_2^2 + \sqrt{(M_{12}^2 + m_{\mathcal{H}}^2 - m_2^2)^2 - 4M_{12}^2(m_{\mathcal{H}}^2 + k_T^2)}}{2M_{12}} \quad \text{and} \quad z^{(2)} = 1 - \frac{m_2^2 + k_T^2}{M_{12}z^{(1)}}. \tag{12}$$

### 2.1.7 Rescue system for anomalies in cluster formation

In some rare cases it may be impossible for gluons to decay or for the produced clusters to decay further or to transition directly into hadrons. Below we outline how the model treats these anomalies:

1. Splitter-spectator system not massive enough:
   If the invariant mass of the splitter–spectator system ($f_1 g_2$ or $g_2 \bar{f}_1$) is not large enough to allow the gluon to split into two constituents,

   $$(p_1 + p_2)^2 < (m_1 + \min_f 2m_f)^2, \tag{13}$$

   with $\min m_f$ the mass of the lightest flavour, the gluon will be removed and its momentum will be added to the spectator momentum ($f_1 g_2 \to f_{1'}$ or $g_2 \bar{f}_1 \to \bar{f}_{1'}$ with $p_1 \to p_1' = p_1 + p_2$).

2. Two-gluon singlet $g_1 g_2$ not massive enough:
   If the invariant mass of a two-gluon system is not large enough to allow splitting one of the gluons,

   $$(p_1 + p_2)^2 < \min_f 4m_f^2, \tag{14}$$

   the singlet is treated as a cluster, and the cluster rescue system discussed below is invoked.

3. Singlet system below minimal hadron mass:
   With the masses of light quarks usually ignored in the parton shower it is possible to arrive at two-quark systems $f_1 \bar{f}_2$ with a mass below the lightest allowed hadron,

   $$(p_1 + p_2)^2 < \min_h m_{h[f_1\bar{f}_2]}^2, \tag{15}$$

usually this implies that $(p_1 + p_2)^2 < (m_{f_1} + m_{\tilde{f}_2})^2$. In this case, SHERPA reshuffles momenta from another singlet system or one of the already produced clusters such that the light system can directly transfer to the lightest allowed hadron.

### 2.1.8 Distributions characterising cluster formation

In Fig. 1 we exhibit two distributions that characterise this initial step of the cluster fragmentation model, namely, firstly, the distribution of primary cluster masses in the left panel, and secondly their multiplicity in the right panel. They have been obtained after the CSShower, with no multijet merging and using the tuned parameters of the cluster fragmentation[1]. The tuned values of the parameters are given in Appendix B.

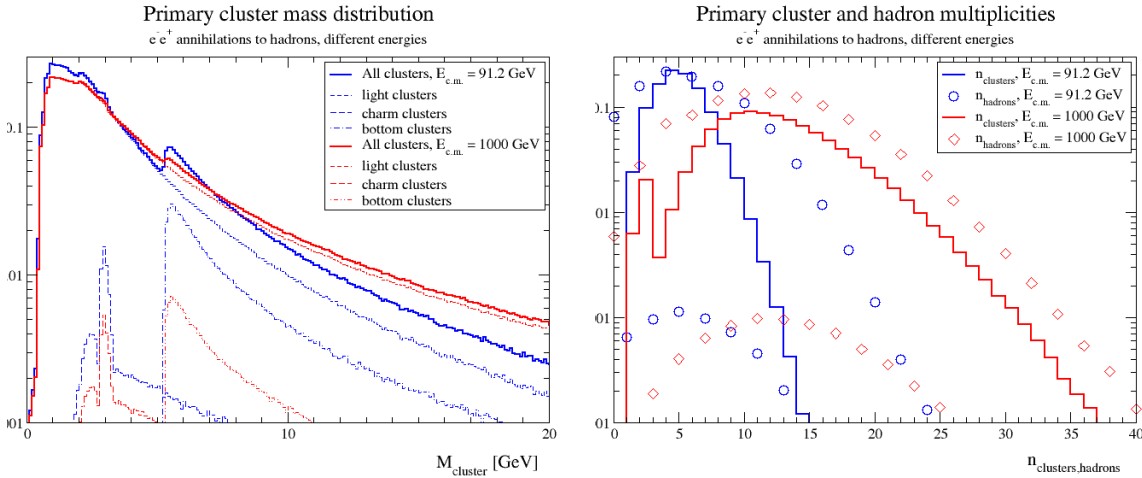

Figure 1: Mass (left panel) and multiplicity distributions (right panel) of primary clusters and hadrons in $e^+e^- \to$ hadrons events at varying centre-of-mass energies.

The cluster mass distribution follows what is expected from the distribution of partons produced in the parton shower, with a peak at about 1 GeV, anticipated from the parton shower cut-off $p_{\perp,0} = 1$ GeV. As there are more and potentially more massive clusters produced at higher energies, this peak is less pronounced at higher energies, compensated by a higher tail of the distribution. In fact, it is entirely possible that, due to its probabilistic nature, the parton shower does not emit a single parton, and, consequently, there would be only a single primary cluster with the full centre-of-mass energy of the $q\bar{q}$ pair as mass. We also observe that mass thresholds of heavy quarks and, more faintly, of diquarks, are visible in the overall mass distribution. While in particular the bottom and less so the charm thresholds are fairly pronounced at the $Z$-pole, $E_{\text{c.m.}} = 91.2$ GeV, they are not as prominent at $E_{\text{c.m.}} = 1000$ GeV. This is due to two effects. First of all, due to their coupling, down-type quarks, including $b$ quarks, are more copiously produced at the $Z$ pole compared to the up-type quarks, thereby explaining the somewhat larger size of the bottom peak compared to the charm-bump at the $Z$ pole. Secondly, the parton shower produces mainly gluons, while the production of heavy quark pairs in gluon splitting is suppressed by their mass. As a consequence, there are proportionally more light flavours and more light clusters produced which suppresses the significance of the heavy quark thresholds.

---

[1] We have also set all heavy mesons and baryons stable in the simulation to suppress the fragmentation in their possible parton-level decays in the simulation.

From the right panel of Fig. 1 we can also see that the number of primary clusters increases from $\langle n_{\text{clusters}} \rangle \approx 5$ in the peak at $E_{\text{c.m.}} = 91.2$ GeV to $\langle n_{\text{clusters}} \rangle \approx 12$ in the peak at $E_{\text{c.m.}} = 1000$ GeV, a very good realisation of logarithmic scaling. We have also shown the number of primary hadrons there that emerge directly from those primary clusters that are not heavy enough to produce secondary clusters. Not unexpected, due to the clusters disintegrating in binary decays, typically we find even hadron numbers. The odd hadron multiplicities, usually at the per-mil level or below, are a consequence of individual clusters transforming directly into hadrons as part of the rescue system.

## 2.2 Cluster fission

If clusters made of two flavours $f_1$ and $\bar{f}_2$ are heavy enough – *i.e.* above their threshold for decays into hadrons, see below – they will decay into secondary clusters, $\mathcal{C}[f_1 \bar{f}_2] \rightarrow \mathcal{C}[f_1 \bar{f}] + \mathcal{C}[f \bar{f}_2]$. In SHERPA this proceeds by first selecting the non-perturbatively produced flavour $f$ associated to the decay, before defining the decay kinematics.

### 2.2.1 Non-perturbative flavour production

Similar to the treatment in the non-perturbative decays of gluons at the end of the parton shower, the produced flavours $f + \bar{f}$ are determined according to the "popping" probabilities $\mathcal{P}_f$. In analogy to the case of gluon splitting above, Eq. (2), the available flavours are only constrained by the condition that

$$M_{\mathcal{C}} - m_{f_1} - m_{\bar{f}_2} > 2m_f \,. \tag{16}$$

The underlying assumption here is that in practically all cases it will be possible to produce hadrons from the resulting $\{f_1 \bar{f}\}$ and $\{f \bar{f}_2\}$ systems, due to the constituent masses being similar to the hadron masses.

It is worth stressing here that, in contrast to early realisations of the cluster hadronisation model, in SHERPA diquarks are allowed to constitute clusters[2]. Allowing diquark production at every stage in the hadronisation process, *i.e.* in both gluon decays and in the subsequent fission of clusters into secondary clusters, softens their strong correlation. This represents an alternative the popcorn mechanism [42, 43] in the Lund string model within cluster hadronisation models, which softens the previous strong correlation of baryon–anti-baryon pairs.

### 2.2.2 Fixing the kinematics

Having fixed the "popped" flavour $f$ and therefore the flavour contents $\{f_1 \bar{f}\}$ and $\{f \bar{f}_2\}$ of the two systems produced in the decay, their kinematics must now to be fixed. This is achieved in the rest frame of the cluster, where the momenta $p_1$ and $p_2$ of particles $f_1$ and $\bar{f}_2$ are oriented parallel to the positive and negative $z$-axis, allowing us to introduce $n_{\pm}^{\mu} = (1, 0, 0, \pm 1)$. In the rest frame of the cluster therefore

$$p_1 = \frac{m_{\mathcal{C}}}{2} \left[ z_+ n_+^{\mu} + (1 - z_-) n_-^{\mu} \right] \quad \text{and} \quad p_2 = \frac{m_{\mathcal{C}}}{2} \left[ (1 - z_+) n_+^{\mu} + z_- n_-^{\mu} \right], \tag{17}$$

with

$$z_{\pm} = \frac{M_{\mathcal{C}}^2 \pm m_{f_1}^2 \mp m_{\bar{f}_2}^2 + \sqrt{(M_{\mathcal{C}}^2 - m_{f_1}^2 - m_{\bar{f}_2}^2)^2 - 4m_{f_1}^2 m_{\bar{f}_2}^2}}{2M_{\mathcal{C}}^2} \,. \tag{18}$$

---

[2]Another alternative, proposed in [44], is to construct baryonic clusters directly from three quarks or three anti-quarks.

Similarly, the four four-momenta $p^\mu_{11}$, $p^\mu_{12}$, $p^\mu_{21}$ and $p^\mu_{22}$ of the four outgoing flavours $f_1$, $\bar{f}$, $f$, and $\bar{f}_2$ are parameterized as

$$
\begin{aligned}
p^\mu_{11} &= \frac{m_C}{2}\Big[ & x^{(1)}z^{(1)}n^\mu_+ &+ & y^{(1)}(1-z^{(2)})n^\mu_- & \Big], \\
p^\mu_{12} &= \frac{m_C}{2}\Big[ & (1-x^{(1)})z^{(1)}n^\mu_+ &+ & (1-y^{(1)})(1-z^{(2)})n^\mu_- & \Big] &+ k^\mu_\perp, \\
p^\mu_{21} &= \frac{m_C}{2}\Big[ & (1-x^{(2)})(1-z^{(1)})n^\mu_+ &+ & (1-y^{(2)})z^{(2)}n^\mu_- & \Big] &- k^\mu_\perp, \\
p^\mu_{22} &= \frac{m_C}{2}\Big[ & x^{(2)}(1-z^{(1)})n^\mu_+ &+ & y^{(2)}z^{(2)}n^\mu_- & \Big],
\end{aligned}
\tag{19}
$$

where

$$
x^{(1,2)} = \frac{\tilde{q}^2_{1,2} + m^2_{f_1,\bar{f}_2} - (m^2_f + k^2_T) \pm \sqrt{[\tilde{q}^2_{1,2} - (m^2_{f_1,\bar{f}_2} - m^2_f + k^2_T)]^2 - 4m^2_{f_1,\bar{f}_2}(m^2_{f_1,\bar{f}_2} + k^2_T)}}{2\tilde{q}^2_{1,2}},
$$

$$
y^{(1,2)} = \frac{1}{x^{(1,2)}} \cdot \frac{m^2_{f_1,\bar{f}_2}}{\tilde{q}^2_i},
\tag{20}
$$

and the masses of the two resulting clusters,

$$
\tilde{q}^2_{1,2} = z^{(1,2)}(1-z^{(2,1)})M^2_C + k^2_T.
\tag{21}
$$

The absolute value $k_T$ of the transverse momentum $k_\perp$, with respect to the axis defined by the momenta of the cluster constituents, is selected according to the same Gaussian as before, Eq. (6), with the same parameter $k_{\perp,0}$. This leaves the longitudinal momenta fractions $z_{1,2}$, or, equivalently, the masses of the outgoing clusters to be determined in order to fix the kinematics of the cluster decay. SHERPA offers two methods to achieve this:

1. Fixing the longitudinal momenta fractions $z_{1,2}$
   The $z^{(i)}$ are selected according to a probability

$$
\mathcal{P}(z) = z^\alpha (1-z)^\beta \exp\left[ -\frac{\gamma}{z} \cdot \frac{k^2_T + (m_{f_1} + m_{\bar{f}_2})^2}{k^2_{\perp,0}} \right],
\tag{22}
$$

a form similar to the Lund symmetric fragmentation function [4], with parameters $\alpha$, $\beta$, and $\gamma$ depending on whether flavour $i$ is a light quark, a heavy quark, or a diquark or whether the decaying cluster contains a beam remnant, a parton stemming from the non-perturbative break-up of incident hadrons at hadron colliders. The $z$ ranges are given by

$$
z^{(1,2)}_{\text{min,max}} = \frac{M^2_C - (M^{(2,1)}_{\text{min}})^2 + (M^{(1,2)}_{\text{min}})^2 \mp \sqrt{\left[ M^2_C - (M^{(1)}_{\text{min}})^2 - (M^{(2)}_{\text{min}})^2 \right]^2 - 4(M^{(1)}_{\text{min}})^2 (M^{(2)}_{\text{min}})^2}}{2M^2_C},
\tag{23}
$$

where the $M^{(i)}_{\text{min}}$ denote the minimal mass of a hadron system that can be produced from the flavour pair $\{f_1\bar{f}\}$ or $\{f\bar{f}_2\}$, i.e., denoting the masses for a hadron $\mathcal{H}$ with flavour content $f\bar{f}'$ as $m_{\mathcal{H}}[f\bar{f}']$,

$$
M^{(1)}_{\text{min}} = \min_{f'} \left( m_{\mathcal{H}_{11}[f_1\bar{f}']} + m_{\mathcal{H}_{12}[f'\bar{f}]} \right) \quad \text{and} \quad M^{(2)}_{\text{min}} = \min_{f'} \left( m_{\mathcal{H}_{21}[f\bar{f}']} + m_{\mathcal{H}_{22}[f'\bar{f}_2]} \right).
\tag{24}
$$

2. Fixing the outgoing cluster masses

   Alternatively, the $z^{(i)}$ can be calculated from the $\tilde{q}_i$, the masses of the two clusters produced in the decay. They are selected according to

$$\tilde{q}_i^2 = \left( M_{\min}^{(i)} + \Delta M^{(i)} \right)^2 . \tag{25}$$

SHERPA offers a number of different, relatively simple options to calculate the $\Delta M$, with (un-normalized) probabilities distributed according to

$$\mathcal{P}(\Delta M) = \begin{cases} \exp\left[ -\frac{\Delta M}{\gamma k_{\perp,0}} \right] & \text{(exponential)} \\ \exp\left[ -\frac{(\Delta M - \langle \Delta M \rangle)^2}{\gamma k_{\perp,0}} \right] & \text{(Gaussian)} \\ \exp\left[ -\frac{(\log \Delta M - \log \langle \Delta M \rangle)^2}{\gamma k_{\perp,0}} \right] & \text{(log-normal)} \end{cases}, \tag{26}$$

where the mean value of the mass shift, $\langle \Delta M \rangle = k_{\perp,0}$. The $z^{(i)}$ are calculated from the two $\tilde{q}_i$ as

$$z^{(1,2)} = \frac{M_C^2 + \tilde{q}_{1,2}^2 - \tilde{q}_{2,1}^2 + \sqrt{(M_C^2 + \tilde{q}_1^2 - \tilde{q}_2^2)^2 - 4M_C^2(\tilde{q}_1^2 + k_T^2)}}{2M_C^2} , \tag{27}$$

and, as before, the azimuthal angle w.r.t. the momenta of the cluster constituents is chosen flat.

### 2.2.3 Secondary clusters directly transitioning into hadrons

Similar to the cluster produced in the non-perturbative gluon splitting, also the masses of secondaries produced in cluster fission may be below the threshold for direct transition to hadrons, *cf.* Eq. (9). Selecting the respective hadron type according to the probability given in Eq. (10), the kinematics of cluster fission to be adjusted to accommodate decays into one hadron plus a cluster or two hadrons only. The two momenta for the secondaries are given by

$$\begin{aligned} p_1^\mu &= M_C\left[ \quad\quad z^{(1)} n_+^\mu \quad + \quad (1-z^{(2)}) n_-^\mu \quad \right] \quad + \quad k_\perp^\mu, \\ p_2^\mu &= M_C\left[ (1-z^{(1)}) n_+^\mu \quad + \quad\quad z^{(2)} n_-^\mu \quad \right] \quad - \quad k_\perp^\mu, \end{aligned} \tag{28}$$

with the $z^{(1,2)}$ given by Eq. (27) where the cluster masses are replaced by hadron masses where necessary. In case this results in one cluster and one hadron, the momenta of flavours constituting the cluster are boosted into the new frame given by the updated cluster momentum.

### 2.2.4 Distributions characterising cluster fission

In Fig. 2 we depict distributions characterising the decay of clusters for different primary clusters: $u\bar{u}$ and $c\bar{c}$ clusters with a mass of 10 GeV and $b\bar{b}$ clusters with a mass of 20 GeV.

We show the mass distribution of secondaries emerging in the first generation of decays and their $x_p = |\vec{p}_c|/|\vec{p}_f|$ distribution, where $p_f$ is the momentum of the quark inside the primary cluster giving rise to them, and $p_c$ are the momenta of the secondaries. As expected the finite constituent quark masses lead to thresholds for the clusters containing them, leading to a minimal cluster mass of about 600 MeV for clusters containing only light quarks, of about 2100 MeV for single-charmed clusters and of about 5300 MeV for cluster containing a bottom quark. Conversely, the $x_p$-distribution shows and increasingly sharp peak for increasing quark masses, for two reasons, both of which can be directly read off the "fragmentation function"

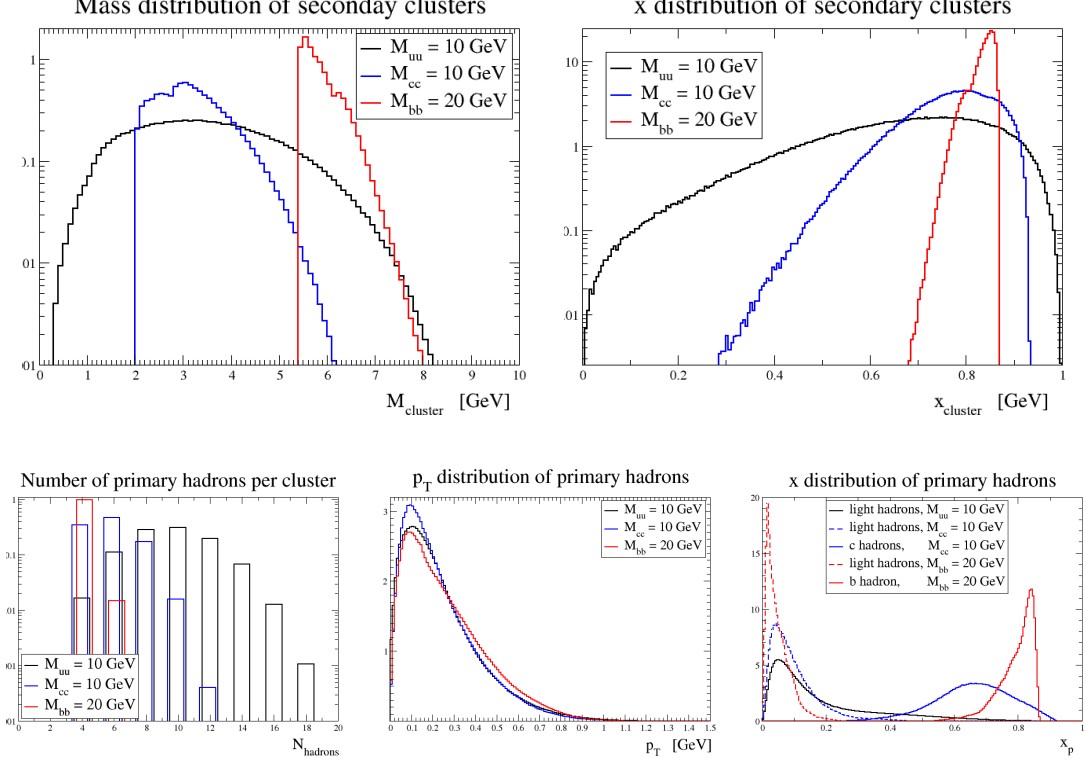

Figure 2: Mass distributions of secondary clusters produced in the decays of clusters of different mass and flavour composition (upper left), the corresponding multiplicity of primary hadrons (lower left), their transverse momenta with respect to the axis defined by the momenta of the original cluster constituents (lower middle), and the longitudinal momentum fractions (lower right). The parameters from Tab. 4, have been used.

in Eq. (22). First of all, comparing the tuned $\alpha$, $\beta$, and $\gamma$ parameters defining the cluster splittings for light and heavy flavours inside the cluster, it is apparent that the heavy-quark "fragmentation" function is much harder than its light-quark counterpart. In addition, while for the two heavy flavours – the $c$ and $b$ quarks – these parameters are identical, small values of $z$ experience a suppression that, up to parameters, scales like $\exp(-m_Q^2)/z$, resulting in a much more pronounced suppression of small $z$ values for the heavier quarks.

In the same figure, we also show the resulting overall multiplicity of primary hadrons in the full decay chain, their transverse momenta w.r.t. the axis defined by the cluster constituents, and their $x_p = |\vec{p}_h|/|\vec{p}_f|$ distribution. We observe that the number of hadrons decreases with increasing mass of the original constituents, as simple result of a combination of available phase space, which is constrained by the masses of the quarks, and the harder cluster splitting for the heavier flavours. This manifests itself also in the $x_p$ distributions where we see that the heavy hadrons carry more of the original quark momentum than their light counterparts, a trend that is also more pronounced for $b$ quarks over $c$ quarks. Finally, we also note that the transverse momentum distributions of the hadrons are nearly uniform for all three cases.

## 2.3 Cluster decays into hadrons

### 2.3.1 Selecting hadrons

Once clusters are produced, either in the cluster formation phase following the parton shower or through fission into secondary clusters, they may decay, if they are light enough, into hadrons $\mathcal{H}_1 + \mathcal{H}_2$. The threshold for the decays into pairs of hadrons is given by the combination of the lightest possible and the heaviest possible masses. For clusters $\mathcal{C}[f_1 \bar{f}_2]$ made from a colour triplet $f_1$ – either a quark or an anti–diquark – and an anti–colour triplet $\bar{f}_2$ – either an anti–quark or a diquark – this would proceed by non-perturbatively producing a flavour–anti-flavour pair $f\bar{f}$ such that

$$\mathcal{C}[f_1 \bar{f}_2] \rightarrow \mathcal{H}_1[f_1 \bar{f}] + \mathcal{H}_2[f \bar{f}_2]. \tag{29}$$

The mass threshold for decays, $M_{\mathrm{dec}}[f_1 \bar{f}_2]$ is defined by

$$M_{\mathrm{dec}}[f_1 \bar{f}_2] = x_{\mathrm{dec}} \min_f \left( m_{\mathcal{H}_1[f_1 \bar{f}]} + m_{\mathcal{H}_2[f \bar{f}_2]} \right) + (1 - x_{\mathrm{dec}}) \max_f \left( m_{\mathcal{H}_1[f_1 \bar{f}]} + m_{\mathcal{H}_2[f \bar{f}_2]} \right), \tag{30}$$

with a tuning parameter $x_{\mathrm{dec}}$. If the cluster mass $M_{\mathcal{C}} < M_{\mathrm{dec}}$ then the cluster will decay into two hadrons or a hadron and a photon.

The exact channel for decays $\mathcal{C} \rightarrow \mathcal{H}_1 + \mathcal{H}_2$ is selected according to the respective weights, given by

$$\mathcal{P}_{\mathcal{C}[f_1 \bar{f}_2] \rightarrow \mathcal{H}_1[f_1 \bar{f}] + \mathcal{H}_2[f \bar{f}_2]} = \mathcal{P}_f \times w_{\mathcal{H}_1} \times \left| \psi_{\mathcal{H}_1}(f_1 \bar{f}) \right|^2 \times w_{\mathcal{H}_2} \times \left| \psi_{\mathcal{H}_2}(f \bar{f}_2) \right|^2$$

$$\times \frac{\sqrt{(M_{\mathcal{C}}^2 - m_{\mathcal{H}_1}^2 - m_{\mathcal{H}_2}^2)^2 - 4 m_{\mathcal{H}_1}^2 m_{\mathcal{H}_2}^2}}{8 \pi M_{\mathcal{C}}^2} \times \left[ \left( \frac{m_{\mathcal{H}_1}}{M_{\mathcal{C}}} \right)^{\chi} + \left( \frac{m_{\mathcal{H}_2}}{M_{\mathcal{C}}} \right)^{\chi} \right], \tag{31}$$

where $\mathcal{P}_f$ again is the "popping" probability for the production of flavour $f$. The $\psi_{\mathcal{H}}$ are the flavour wave functions of the hadrons, cf. Appendix A, and the $w_{\mathcal{H}}$ are additional weights to produce hadron $\mathcal{H}$, composed as products of the multiplet weight and, possibly, additional, hadron-type specific weights, listed in Tab. 5. $\chi$ is an additional tunable parameter, which by default has been chosen to vanish, $\chi = 0$.

### 2.3.2 Rescue system for light clusters

In SHERPA's model, it is possible that clusters are created that are too light to decay into hadrons. An example for this is the possible creation of clusters consisting of two diquarks, with a mass below the two-baryon threshold. To avoid having to repeat possible costly parts of the event generation necessitates an extension of the cluster decay model to capture these cases:

1. Clusters made of two diquarks: $\mathcal{C}[(ij)(\bar{k}\bar{l})]$ with $M_{\mathcal{C}} < m_{B[(ij)]} + m_{B[(\bar{k}\bar{l})]}$

   The most obvious case are clusters that consist of two diquarks $(ij)$ and $(\bar{k}\bar{l})$ with a mass that is below the mass threshold of baryon–anti-baryon pairs containing them. In this case, SHERPA splits the two diquarks and creates two quark–anti-quark pairs from them, with random pairing, i.e. $\{i\bar{k}\} + \{j\bar{l}\}$ or $\{i\bar{l}\} + \{j\bar{k}\}$. Weights for kinematically allowed decays of the cluster into two mesons are calculated according to Eq. 31, and one decay mode is selected according to them:

$$\mathcal{C}[(ij)(\bar{k}\bar{l})] \rightarrow \mathcal{M}[i\bar{k}] + \mathcal{M}[j\bar{l}] \quad \text{or} \quad \mathcal{M}[i\bar{l}] + \mathcal{M}[j\bar{k}]. \tag{32}$$

   If there is, however, no allowed decay of the cluster into two mesons, SHERPA will try to "annihilate" a flavour pair. For example, if $i = k$ in the two diquark constituents, they

will be assumed to have "cancelled" each other out. The cluster will then decay into a photon and a meson with the remaining flavour quantum numbers $\{j\bar{l}\}$:

$$\mathcal{C}[(ij)(\bar{i}\bar{l})] \rightarrow \mathcal{M}[j\bar{l}] + \gamma\,, \tag{33}$$

where the meson is selected according to the available phase space for the decay, if more than one decay channel is kinematically allowed.

2. Clusters not made of two diquarks: $\mathcal{C}[f_1\bar{f}_2]$ with $M_{\mathcal{C}} < m_{M[f_1\bar{f}_2]}$
   Clusters too light to decay into two hadrons will decay radiatively into a hadron and a photon, where the hadron is selected according to weights given by

$$\mathcal{P}_{\mathcal{C}[f_1\bar{f}_2]\rightarrow\mathcal{H}_1[f_1\bar{f}_2]+\gamma} = w_{\mathcal{H}}\left|\psi_{\mathcal{H}}(f_1\bar{f}_2)\right|^2 \times \frac{M_{\mathcal{C}}^2 - m_{\mathcal{H}}}{8\pi M_{\mathcal{C}}^2} \times \left(\frac{m_{\mathcal{H}}}{M_{\mathcal{C}}}\right)^{\chi}\,. \tag{34}$$

3. Clusters made of $q\bar{q}$ pairs: $\mathcal{C}[q\bar{q}]$ with $M_{\mathcal{C}} < m_{M[q\bar{q}]}$
   This assumes that the cluster cannot decay into a pair of hadrons containing the quark and the anti-quark or the lightest meson made of the $q\bar{q}$-pair and a photon. Then the model annihilates the $q\bar{q}$-pair to give rise to pions or photons, namely:

   - if $M_{\mathcal{C}} < M_{\pi\gamma}$, the threshold for $\mathcal{C} \rightarrow \pi^0\gamma$, the cluster will decay into two photons:

$$\mathcal{C}[q\bar{q}] \rightarrow \gamma + \gamma\,; \tag{35}$$

   - if $M_{\pi\gamma} < M_{\mathcal{C}} < M_{\pi\pi}$, the threshold for $\mathcal{C} \rightarrow \pi\pi$, the cluster will decay to a $\pi^0$ and a photon:

$$\mathcal{C}[q\bar{q}] \rightarrow \pi^0 + \gamma\,; \tag{36}$$

   - if $M_{\pi\pi} < M_{\mathcal{C}} < M_{\mathcal{M}[q]} + M_{\mathcal{M}[\bar{q}]}$, the cluster will decay to two pions, either $\pi^+\pi^-$, with a probability of 2/3 or $\pi^0\pi^0$ with a probability of 1/3:

$$\mathcal{C}[q\bar{q}] \rightarrow \pi^+ + \pi^- \quad \text{or} \quad \mathcal{C}[q\bar{q}] \rightarrow \pi^0 + \pi^0\,. \tag{37}$$

The treatment outlined above also appiles to colour-singlets made of two gluons that do not have enough mass to decay into constituent quarks.

The two mass thresholds are listed in Tab. 6.

### 2.3.3 Kinematics for cluster decays into hadrons

Having fixed the flavours of the particles that are being produced in the cluster decay, the kinematics is easily constructed in the cluster's rest frame. The transverse momentum $k_\perp$ of the hadrons with respect to the cluster constituents is selected according to the same Gaussian distribution used in gluon decays and cluster fission, Eq. (6), with the same parameter $k_{\perp,0}$, and with a flat azimuthal angle. In contrast to the case of cluster fission, where the masses for the resulting clusters need to be fixed, this completely determines the kinematics of the decay: the longitudinal momenta of the hadrons are easily calculated now from their masses and transverse momenta, and they are aligned with the constituents giving rise to them.

# 3 Colour Reconnections

In SHERPA a simple model for non-perturbative colour reconnections has been made available; it is however at the moment switched off by default. Such soft colour reconnections have first been analysed and modelled in the context of $W$-mass measurements at LEP [28] and about 15 years later in the context of top-mass measurements [29]; they also played an important part in the hadronisation of the final states in multiple–parton interactions in [30]. They can be thought of as resulting from two effects. First of all, parton showers implicitly assume the limit of an infinite number of colours, $N_c \to \infty$, which results in planar colour flows [38] and, therefore, a direct one–to–one connection of quarks and anti-quarks and therefore a unique way in which the first primary clusters are formed. For the actual $N_c = 3$ of QCD, this unique connection of colours and anti-colours is obviously not correct, and one would expect small changes to it. Secondly, in particular in hadron–hadron collisions, and in the presence of multiple parton–parton interactions, one can expect that some of the parton cascades overlap in space–time, increasing the probability for the soft exchange of colours.

In contrast to, *e.g.*, the model implemented in HERWIG [33], and more in line with its realisation in PYTHIA [32], the SHERPA colour reconnection model is invoked at the end of the parton showering step, before the gluons decay and primordial clusters are formed. The model assumes a parton shower in the large $N_c$ limit, where each colour is compensated by exact one anti-colour, and, consequently, the result containing $N$ such colour pairs. SHERPA then repeatedly – $N^2$ times – compares the distances of original and swapped pairs of partons. The distance of two partons $i$ and $j$ is given by

$$d_{ij} = \frac{1}{C} \times \Delta P_{ij} \times \Delta R_{ij}, \tag{38}$$

where $C = 1$ for colour-connected pairs $ij$ and $C = \kappa_C$ for (swapped) pairs. The distances $\Delta P_{ij}$ and $\Delta R_{ij}$ of the two partons in momentum space and the transverse position space are given by

$$\Delta P_{ij} = \begin{cases} \log\left[\dfrac{(p_i + p_j)^2 - (p_i^2 + p_j^2) + Q_0^2}{Q_0^2}\right] & \text{(logarithmic)} \\[2ex] \left[\dfrac{(p_i + p_j)^2 - (p_i^2 + p_j^2) + Q_0^2}{Q_0^2}\right]^{\eta_P} & \text{(power)} \end{cases}, \tag{39}$$

and

$$\Delta R_{ij} = \begin{cases} \left(\dfrac{|x_\perp^{(i)} - x_\perp^{(j)}|^2}{R_0^2}\right)^{\eta_R} & \text{for } |x_\perp^{(i)} - x_\perp^{(j)}|^2 > R_0^2 \\[2ex] 1 & \text{else}, \end{cases}, \tag{40}$$

respectively. Note that spatial distances are relevant only in cases like, for example, hadron–hadron collisions, where the individual scatters of the underlying event can occur at different positions in the transverse plane. Note that if parton $i$ or $j$ is a gluon, the model assumes that its momentum splits equally between the colour and the anti-colour and the corresponding momentum is multiplied by $1/2$ in Eq. (39).

From these distances, the model constructs a "swapping probability", to reconnect parton pairs $il$ and $kj$ rather than the original $ij$ and $kl$, as

$$\mathcal{P}_{\text{swap}} = \frac{\exp\left[-(d_{il} + d_{kj})/\bar{d}\right]}{\exp\left[-(d_{ij} + d_{kl})/\bar{d}\right]}, \tag{41}$$

where the normalisation $\bar{d}$ is given by a sum of distances over all colour pairs,

$$\bar{d} = \frac{1}{N^\kappa} \sum_{\text{pairs\{nm\}}} d_{nm} \,. \tag{42}$$

# 4 Results for $e^-e^+ \rightarrow$ hadrons at $E_{\text{cms}} = 91.2$ GeV

In the following we will show a wide range of data, highlighting various aspects of the SHERPA simulation of QCD events in electron-positron annihilations at the $Z$ pole. The results are obtained after tuning of the model with the PROFESSOR tool [45], and more details on the tuning parameters are discussed in Appendix B. PROFESSOR has been applied after some rough fitting of initial parameters, by oscillating between

- inclusive observables – mainly charged multiplicity, event shapes such as thrust, thrust-major and thrust-minor, and the $b$-fragmentation function. They are most sensitive to the kinematics of gluon and cluster decays: $k_{\perp,0}^2$ and the respective $\alpha$, $\beta$, and $\gamma$.

- individual hadron yields that are most sensitive to the flavour popping parameters $P_f$, hadron–dependent threshold parameter $x_{\text{dec}}$ and modifier $\chi$, and hadron multiplet modifiers $w_\mathcal{H}$. In tunes including the colour reconnection model, its parameters were included in this step, but the overall impact of this additional part of the hadronisation modelling is – as expected – typically quite small in $e^-e^+$ annihilations.

We will start from relatively inclusive observables, such as total hadron multiplicities and their distribution in phase space before focusing on increasingly differential observables, from event shapes over jet distributions and fragmentation functions of individual hadron species to the correlation of identified particles in phase space. By and large, the description of data by SHERPA is satisfactory for the bulk of the events. In these dominant regions of phase space, simulation and data typically agree within experimental uncertainties or at a level of 1-10%.

## 4.1 Inclusive particle distributions

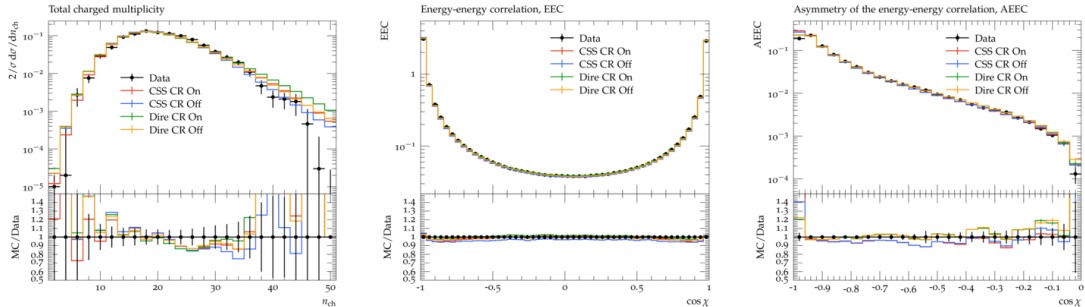

Figure 3: Multiplicities of charged hadrons (left), the energy-energy correlation (middle) and its asymmetry (right) in $e^-e^+ \rightarrow$ hadrons at centre-of-mass energies of 91.2 GeV. The SHERPA results are compared with data from ALEPH [46] for the first, and from DELPHI [47] for the latter two.

We will begin our discussion with the inclusive characteristics of hadron production in $e^-e^+$ annihilations. In Fig. 3 we compare results from SHERPA with corresponding experimental data for charged hadrons multiplicities (from ALEPH [46]), the energy-energy correlation as a

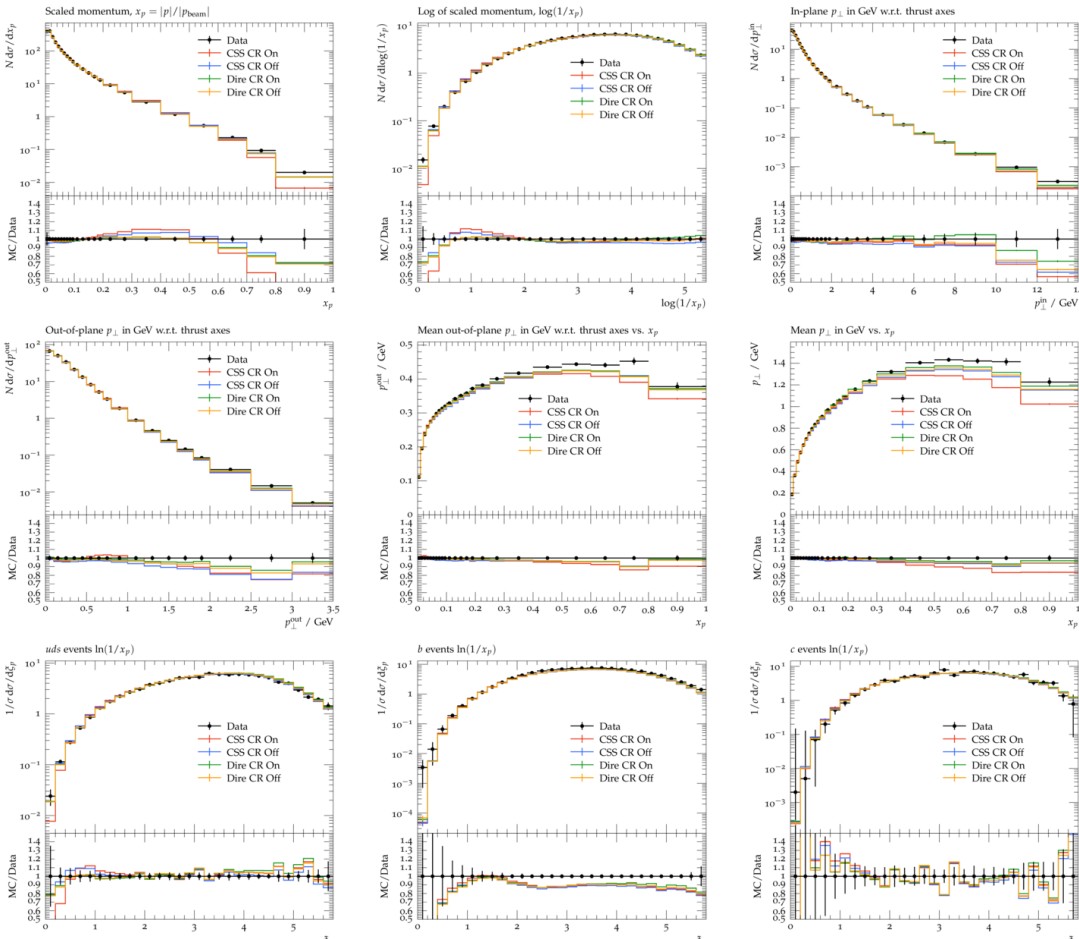

Figure 4: Longitudinal momenta fractions $x_p$ (upper left) and $\log 1/x_p$ (upper middle), transverse momenta in-plane (upper right) and out-of-plane (centre left), with respect to the thrust axis, and the mean value of the out-of-plane and total transverse momenta as a function of $x_p$ (centre middle and right). The SHERPA results are compared with data from DELPHI [47]. Measurement of scaled momentum distributions, $\ln 1/x_p$, for $uds$ quarks (bottom left), $b$ quarks (bottom middle) and $c$ quarks are compared with data from OPAL [48]

function of the angle $\chi$ and its asymmetry (both from DELPHI [47]). By and large, SHERPA is in satisfying agreement with data. However, there are visible deviations in the high-multiplicity tail $N_{ch} \gtrsim 40$ of the charged hadron multiplicity distribution, where SHERPA results fall outside the experimental uncertainties and overshoot data quite significantly. As expected, this is more amplified for tunes where colour reconnections have been switched on, as they often lead to the creation of relatively heavy clusters, which in turn generate a larger hadron multiplicity in their decays.

In Fig. 4 we exhibit the in-plane and out-of-plane $p_\perp$ distributions with the planes defined w.r.t. the thrust axis, and the $x_P$ and $\log 1/x_P$ distributions, and compare the results of the SHERPA simulation with those taken by the DELPHI collaboration in [47]. We compared the measurement of scaled momentum distributions, $\ln 1/x_p$, for quarks with data from OPAL [48]. We observe satisfying overall agreement with data, which is however somewhat hampered by two correlated trends: the cluster fragmentation appears to somewhat overshoot, by about 5-10% the production of hadrons at $x_p$ values of about 0.4 while undershooting at higher

values of $x_p \to 1$. This trend can be further analysed by looking at $-\log 1/x_P = \xi_P$ spectra in events where the $Z$ boson decays into light quarks, charm, or bottom quarks. We find that the overshoot at $x_p \approx 0.1$ is a feature common to all three event categories, and probably most pronounced for the charm-initiated ones. In these latter events, all our four tunes seem to undershoot hadron production by about 10%, and fit data only for the regime around $x_p \approx 0.1$. We also observe that the undershoot for large values of the $x_p$ spectrum is mainly due to the *uds* and *c* event categories – the *b*–initiated events exhibit quite satisfying agreement with data within their uncertainties. Anyway, this undershoot for large $x_p$ is related to the fact that in most cases, clusters decay into two hadrons which then share the momentum quite equally – a typical effect observed in cluster hadronisation models.

## 4.2 Event shapes

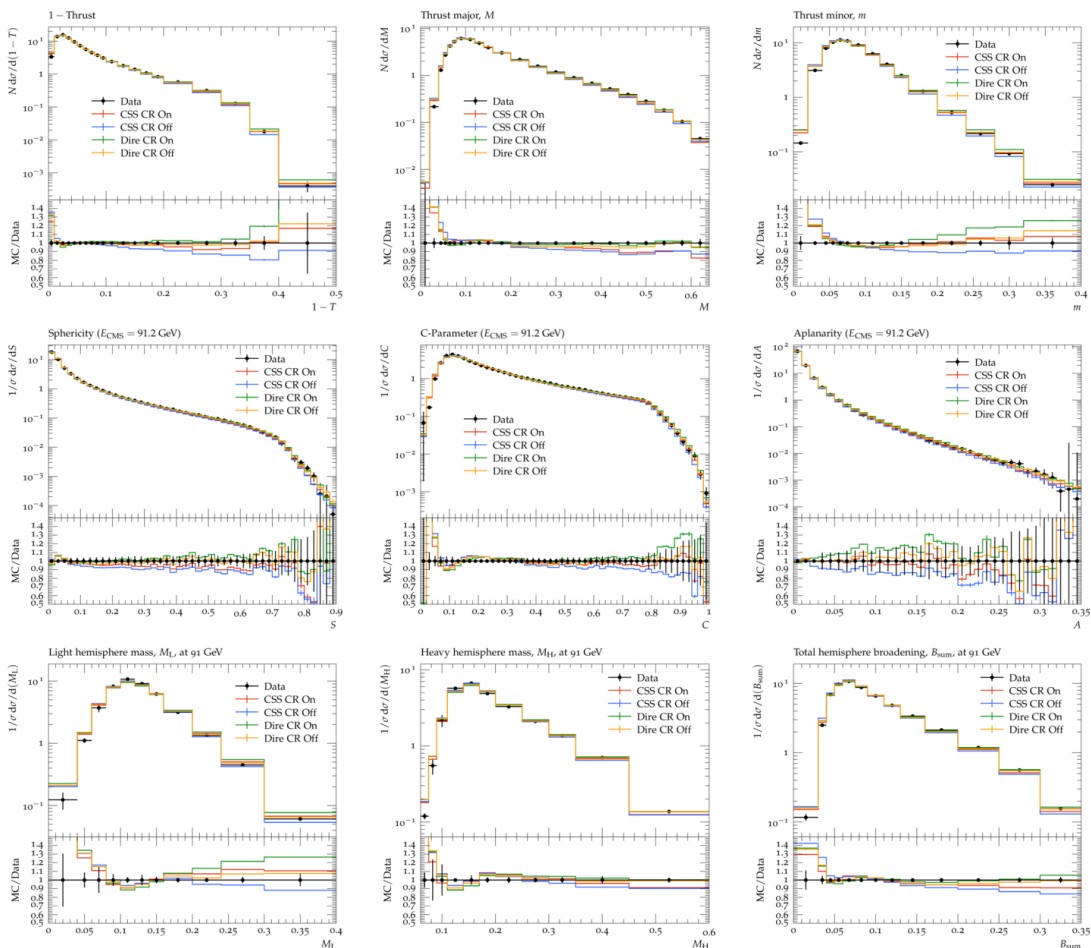

Figure 5: Various event shapes distributions in $e^-e^+ \to$ hadrons at centre-of-mass energies of 91.2 GeV. Upper row, from left to right: thrust, thrust major, thrust minor, all from DELPHI [47]; middle row, from left to right: sphericity, $C$ parameter and aplanarity, all from ALEPH [49]; lower row, from left to right: light and heavy hemisphere mass and total hemisphere broadening, all from OPAL [50].

In Fig. 5 we compare SHERPA results for a number of event shapes with the experimental data. In the upper row we depict thrust $T$ (or more precisely $1-T$), thrust major $M$, and thrust minor $m$, with data taken by DELPHI in [47]. Apart from a significant overshoot in the bins of

small $1 - T$, $M$, and $m$, *i.e.* for extremely pencil-like events, the agreement of the simulation with data is excellent, at the level of 5% or less over a wide range of phase space. This pattern of good to excellent agreement with data, at the 5% or below level, with some overshoots in the extremely pencil-like regime of event topologies, repeats itself also in sphericity $S$, $C$-parameter and aplanarity $A$, displayed in the middle row of Fig. 5, where we use data from ALEPH [49].

In particular the description of the smooth transition from the three to the four-jet regime in the $C$ parameter data at $C \approx 0.75$ is quite impressive. In the lower row of Fig. 5 we display the light and heavy hemisphere masses, $M_L$ and $M_H$, as well as the total hemisphere broadening, $B_{\text{sum}}$ with data from OPAL [50]. Here we observe a difference between data and simulation, as SHERPA undershoots the peak region in $M_L$ and, consequently overshoots the regions of small $M_L \to 0$ and large $M_L$. It is worth noting that the results of this comparison, in particular for $T$, $M$, and $m$, favour the use of DIRE with colour reconnections switched off as the best option, while for the CSShower the inclusion of colour reconnections seems to slightly improve agreement with data to a level not dissimilar to the best option.

## 4.3 Jet distributions

Turning to the description of jets in the electron-positron annihilation events, we focus on differential jet rates in the Durham scheme. They provide an excellent way to judge the performance of the combined parton shower and hadronisation model and its ability to capture QCD dynamics across all scales, from the perturbative to the non-perturbative regime. In Fig. 6 we compare the results of SHERPA with data from a combined JADE plus OPAL analysis [51], and we find, again, excellent agreement of both within the experimental uncertainties. It is worth noting that, again, DIRE without colour reconnections provides the best description of data overall, while the $y_{45}$ distribution for large jet resolutions disfavours the use of DIRE with colour reconnections switched on.

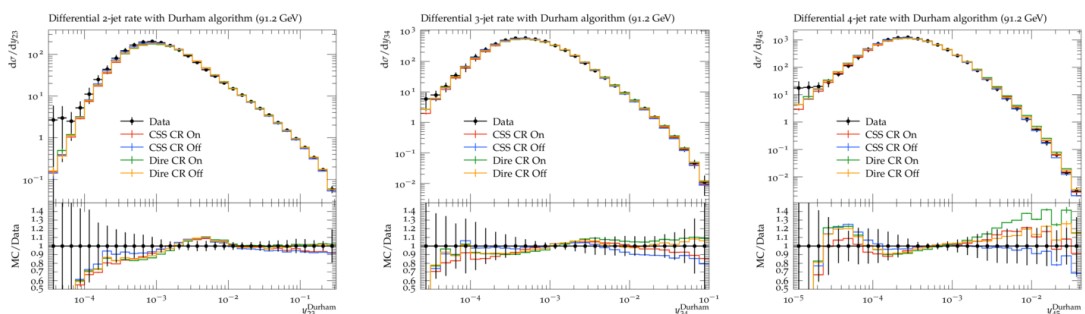

Figure 6: Differential jet rates in the Durham scheme, in $e^- e^+ \to$ hadrons at centre-of-mass energies of 91.2 GeV. SHERPA results are compared with data from a combined JADE-OPAL publication [51], for (from left to right) the differential $2 \to 3$, $3 \to 4$, and $4 \to 5$ jet rates.

## 4.4 Particle production yields and fragmentation functions

Yields for various mesons and baryons are displayed in Fig. 7 and show broad agreement with data and among the four tunes. There is, however, tendency in SHERPA to miss the yields of $\eta$ and $\eta'$ pseudoscalar mesons and of the vector mesons, which our tunes seem to overshoot. Similarly, the production of charmonia states and of some of the excited heavy mesons appears

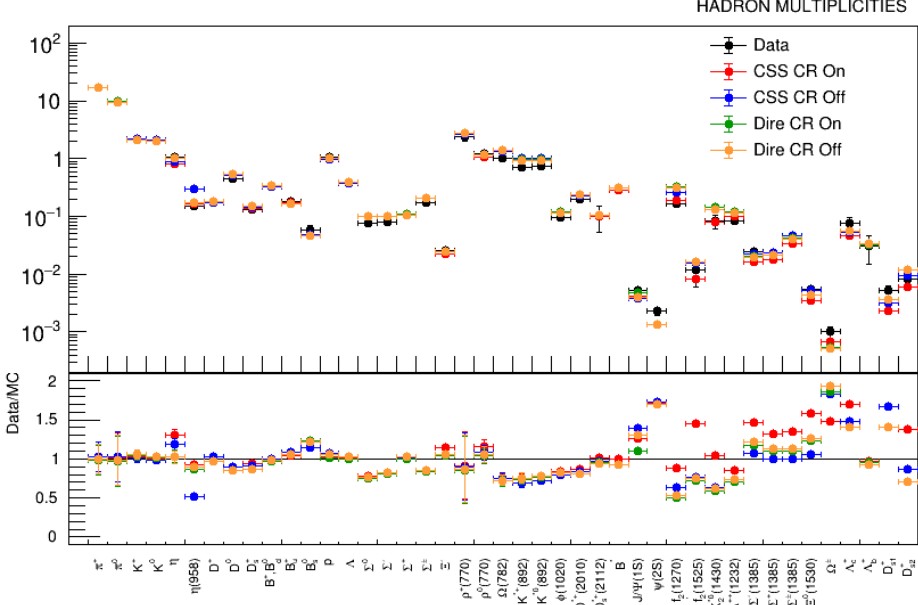

Figure 7: Hadron yields for various species are compared to the data from the particle data group [52], and $\Omega^{\pm}$ is compared to the data from ALEPH [53].

to be not perfect. It could be argued that the former may be alleviated by including the production of such states directly in the parton shower, e.g. by including splitting functions like $g \to J/\psi g$, which emerge from the convolution of the production of $c\bar{c}$ pairs (by sequentially splitting for example $g \to c\bar{c} \otimes c \to cg$) and wave functions that describe their transition to such charmonia states. Turning to baryons we observe good agreement of the tunes with the data for the production of protons and $\Lambda$'s, some overshoot for the other octet baryons (like, the $\Sigma$'s and Cascade baryons) and an undershoot for the decuplet baryons. Especially the latter exhibit a large sensitivity to the overall tune, with some tunes performing (CSShower with colour reconnections disabled) notably better than others.

In Fig. 8 we show a range of $x_p$ distributions for various mesons and baryons. By and large, in the region $x_p \lesssim 0.4$ data and simulation are in excellent agreement with each other and the simulation results rarely fall outside the experimental uncertainties, with the possible exception of some meson distributions overshooting data for very small values of $x_p$. This, of course, could also have been anticipated from the overall quality in the description of the more inclusive data in Fig. 4. It is, however, important to note that our cluster fragmentation model does not arrive at this satisfying result by cross compensating the potentially wrong behaviour of different hadron species but rather arrives at a uniformly good description of hadron production across the board. The only notable exceptions to this picture are the $\Omega$ meson, overshooting data by 40% or more throughout, and the $\phi$ meson, which exhibits a shape difference w.r.t. the data.

In Fig. 9 we show similar results, namely $x_p$ distributions for charged pions $\pi^{\pm}$, kaons $K^{\pm}$ and for protons, in events where the hadrons are produced from $Z$ boson decays into light quarks, charm, or bottom pairs. Again, the simulation agrees quite well with data, with the notable exceptions of a sizable overproduction of pions in $uds$ events at $x_p$ values in the range between 0.2 and 0.5, and a pronounced overproduction of protons in $b$ events at $x_p \approx 0.1$.

Good agreement with data is also found for the modelling of the heavy quark fragmentation process, displayed in Fig. 10. In its left panel we show the $x_E$ distribution of $D^{*\pm}$-mesons, which



Figure 8: From left to right, first row: $x_p$ distributions for $\pi^0$ from (DELPHI [54]), $\pi^\pm$ and $K^\pm$ from DELPHI [55]; second row: $K^0$ from OPAL [56]), $K^{*0}$ from DELPHI [57], $\phi$ from SLD [58] ; third row: $\omega$ and $\eta$ from ALEPH [59], $\Xi^-$ from DELPHI [60]; fourth row: $p, \bar{p}$ from DELPHI [55], $\Sigma^-$ from DELPHI [61] and $x_E$ for $\Sigma^+$ from OPAL [62]

exhibits a tendency of overshooting data by a constant factor, in agreement with the slight overall over-production. In the right panel of this figure we display the $b$-quark fragmentation function which agrees to better than 10% with data when using the DIRE shower, but shows some tension with data when the CSShower is being used. Note that we chose to display the data from SLD [64] which appears to sit in the middle of two other distributions, from ALEPH [65] and OPAL [66], thereby providing some compromise.

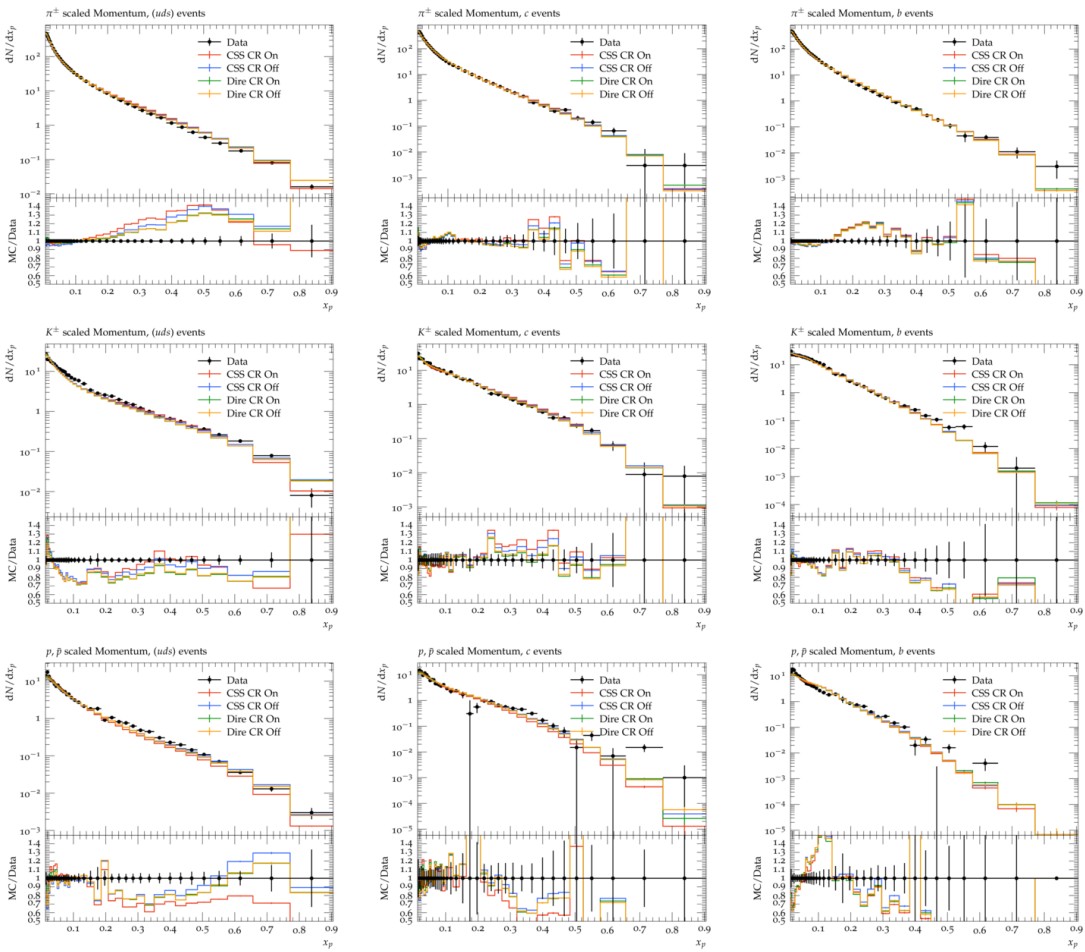

Figure 9: Production of $\pi^\pm$ (upper row), $K^\pm$ (middle row) and $p$ (lower row) in $uds$ (left column), $c$ (middle column), and $b$ (right column) events. All distributions are compared with data from SLD [63].

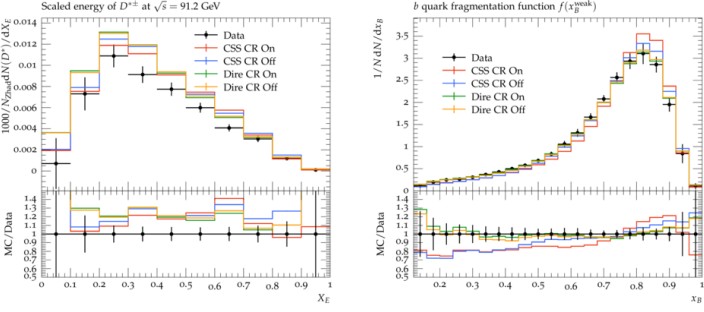

Figure 10: Fragmentation functions for $D^{*\pm}$ (left) and $B$ mesons (right), from OPAL [67] and SLD [64].

## 4.5 Particle correlations

Finally we look at the correlations in baryon production, and in particular the correlation of $\Lambda\bar{\Lambda}$ pairs. This correlation has triggered the development of the popcorn mechanism in the Lund model [43], which softened the relatively strong correlation in baryon production. This

strong coupling of baryons in phase space is due to the fact that in the break-up of the string, and, of course also in the decay of clusters into hadrons, baryon production is associated with the production of a diquark pair, which due to the relatively low scales involved is close in phase space. In the popcorn mechanism this is resolved by "inserting" mesons between the diquarks, but due to the more local nature of cluster fragmentation such a feature is not entirely trivial to encode in the model. In our cluster model, this is resolved by allowing already the gluons to decay into diquark pairs, an option that is usually not available in the version of the cluster fragmentation model where such decays are prohibited by assuming the relatively small non-perturbative gluon "constituent" mass. In SHERPA this limitation is absent, because we assume massless gluons throughout, and generate the mass and momenta of the produced flavours by reshuffling four-momentum from the colour-connected spectator. This allows to generate clusters with the quantum numbers of baryons, which ultimately leads to a drastically reduced correlation of the produced baryons. It is very satisfying to observe that this mechanism apparently resolves the problem of overly correlated baryon-baryon production, *cf.* Fig. 11, where we contrast SHERPA results with data from OPAL [48].

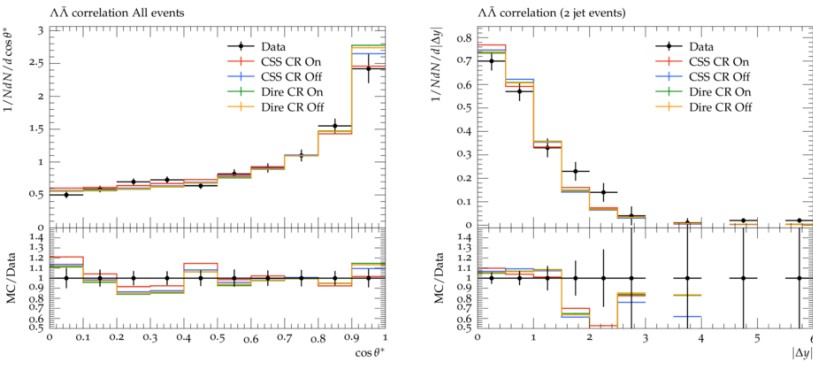

Figure 11: Correlation of $\Lambda\bar{\Lambda}$ pairs in $e^-e^+ \to$ hadrons at 91.2 GeV. We contrast SHERPA results with data from OPAL [48].

## 5 Energy extrapolation

A first step to verify the universality of the hadronisation model is to check the energy extrapolation from the c.m. energy where the model was tuned to data to other energies. The high quality, variety and significance of the results at the $Z$ pole, $E_{\mathrm{c.m.}} = 91.2$ GeV, suggested to use LEP 1 and SLD data for the tuning, and to use data from other energies for some *a posteriori* checks. In the following we will compare the SHERPA results with various observables measured in electron–positron annihilations at $E_{\mathrm{c.m.}} = 14$ GeV, 35 GeV, 44 GeV and 58 GeV.

### 5.1 Results at $E_{\mathrm{c.m.}} = 14$ GeV

Starting at the low energy of $E_{\mathrm{c.m.}} = 14$ GeV, we compare results from the SHERPA simulation with data taken mainly by the TASSO collaboration at the PETRA collider.

In Fig. 12 we focus on inclusive quantities, namely the total charged multiplicity and the energy–energy correlation, comparing our four tunes with data from TASSO [68, 69]. We observe that SHERPA tends to not correctly describe the shape of the charged multiplicity distribution undershooting both the low– and the high–multiplicity region by about 30%. Looking at the energy–energy correlation, we see that this can be traced to some overshoot for large angles, *i.e.* for $\cos\chi$ away from the extreme forward and backward regions. It is however

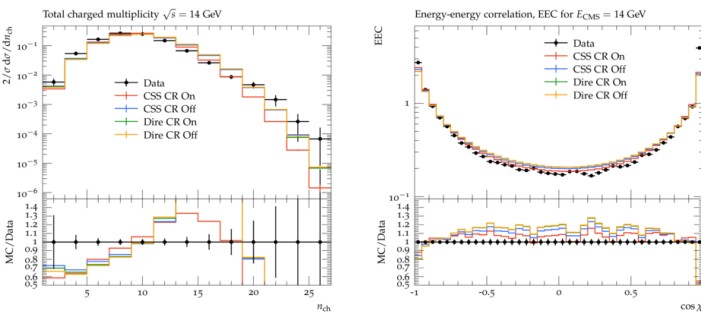

Figure 12: Charged hadron multiplicity from TASSO [68] and energy-energy correlation measured by TASSO [69].

probably worth noting here that the data for the latter look a bit more "bumpy" than, for example, the same observable measured at LEP, and it appears as if the size of the bumps exceeds the experimental uncertainty estimates.

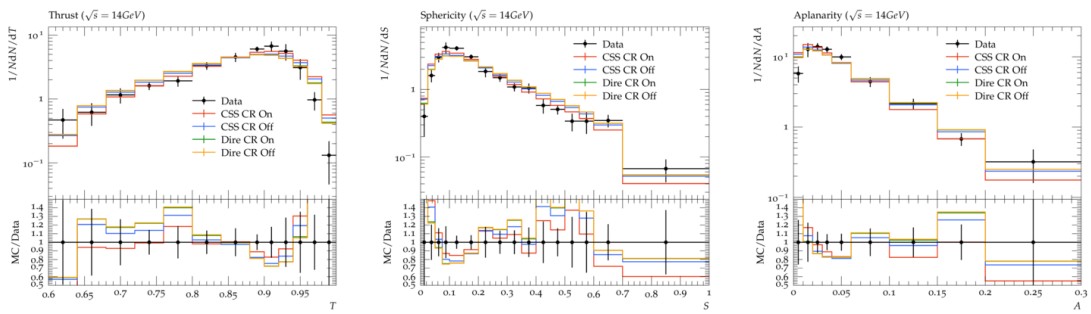

Figure 13: Various event shapes – from left: thrust, sphericity, and aplanarity measured by TASSO [70].

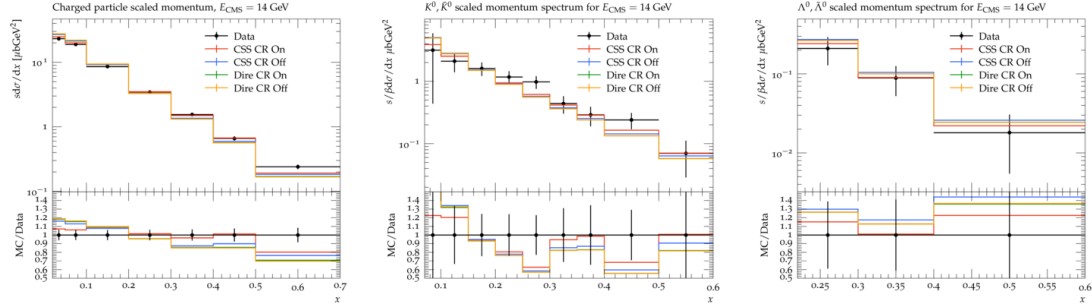

Figure 14: Momenta fractions of charged hadrons from TASSO [71], and of $K^0$ mesons, $\Lambda^0$ baryons, measured by TASSO [72].

We continue by comparing the results for some event shapes – in particular thrust, sphericity, and aplanarity, in Fig. 13, again all taken by TASSO in [70]. Apart from the region of extremely pencil-like events, which SHERPA overestimates significantly, the simulation agrees with experimental data within their uncertainties, indicating that apart from hadron multiplicities the simulation captures overall event characteristics satisfactorily.

In Fig. 14 we exhibit the comparison some particle spectra data from [71,72]. The SHERPA results for the momenta spectra of charged particles, which of course are dominated by the $\pi^{\pm}$, exhibit a slight tilt towards the softer end of the spectrum. At $x$-values in the region of $x \lesssim 0.15$, or momenta of the order of about 1 GeV or below, the fragmentation model tends to overproduce the particle yields by up to about 20%, depending on the tune. A similar behaviour also appears in the neutral kaon spectra: although they exhibit larger uncertainties, allowing SHERPA to agree with data within their uncertainties, the central values are in similar disagreement. Surprisingly enough this is not the case for the $\Lambda$ baryons, which agree well with data.

## 5.2 Results at $E_{\mathrm{c.m.}} = 35$ GeV

In Fig. 15 we depict the event shapes thrust and sphericity, as well as the scaled momentum distribution, measured by TASSO [73] at a centre-of-mass energy of 35 GeV. There is a common trend in the event shape observables: all SHERPA tunes tend to overshoot the bins corresponding to more pencil like events, at $T \approx 1$ and $S \approx 0$, and $A \approx 0$, while somewhat undershooting the peak region by about 10-15%. On the other hand, the agreement in the $x_p$ distribution is quite satisfying, apart from the large $x_p$ region, $x_p \geq 0.5$, where all tunes undershoot the data. This appears to be one of the usual feature of cluster hadronisation models, related to the fact that the clusters tend to decay too democratically.

The agreement of simulation and data in the scaled momentum spectrum of charged particles is also reflected in corresponding spectra for individual neutral mesons, *cf.* Fig 16, where we show the $x$ distribution $K^0$ mesons from CELLO [74] and of $\eta$ and $\rho^0$ mesons from JADE [75, 76].

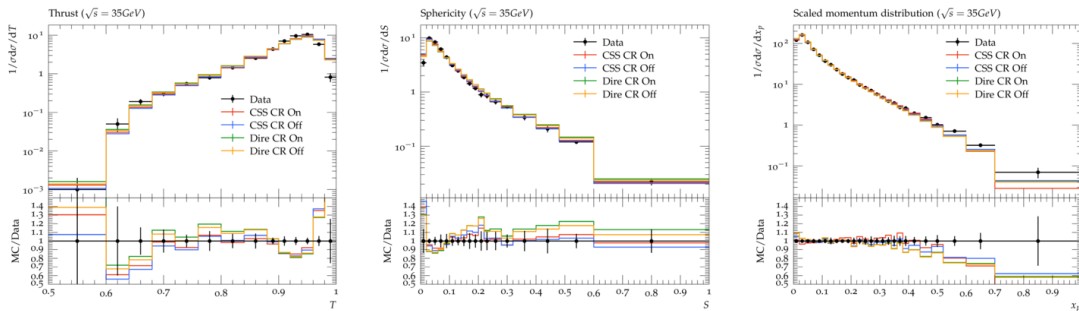

Figure 15: Various event shapes – from left: thrust, sphericity, and $x_p$ distribution, all measured by TASSO [73].

## 5.3 Results at $E_{\mathrm{c.m.}} = 44$ GeV

In Fig. 17 we compare results obtained with the four SHERPA tunes with a set of different observables. We observe that apart from the most pencil-like bin at $T \approx 1$, the simulation describes the thrust distribution measured by TASSO within the experimental uncertainties. This is also true for the differential 2-jet rate in the Durham scheme, $y_{23}$, where SHERPA satisfyingly reproduces the JADE data [51]. The pattern repeats itself with a significantly different observable, the scaled momentum distribution of (anti-)protons, where, again, agreement of simulation with data, again from TASSO [70,77], is quite good, with maybe a little bit of a relative shape difference.

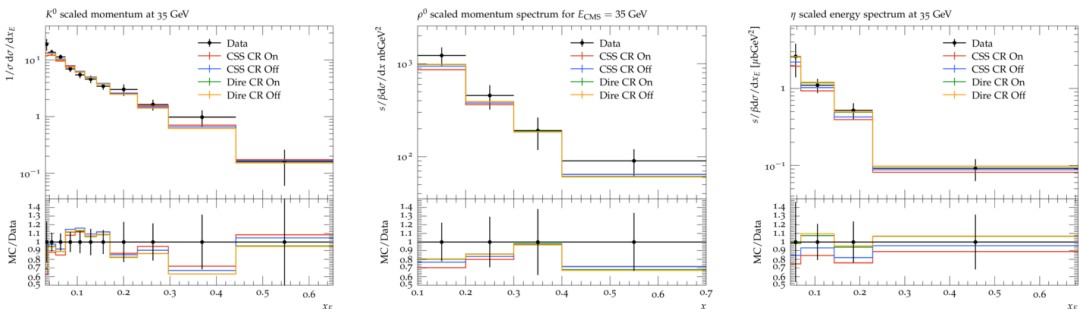

Figure 16: Scaled momentum of $K^0$ mesons (left) by CELLO [74], and of $\rho^0$ (centre) and $\eta$ (right), both by JADE [75,76].

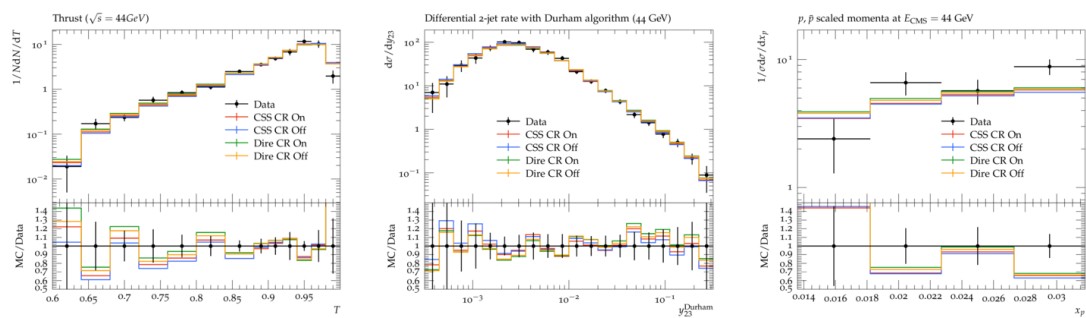

Figure 17: SHERPA results for thrust (left), the differential 2-jet rate in the Durham scheme (middle) and $p, \bar{p}$ scaled momenta (right) compared to results from TASSO [70], JADE-OPAL [51], and TASSO [77], respectively.

## 5.4 Results at $E_{\text{c.m.}} =$ 55 GeV, 58 GeV and 59.5 GeV

Finally turning to centre-of-mass energies of 55-58 GeV, we compare SHERPA results to data from AMY and TOPAZ. In Fig. 18 we look at some event shapes like thrust and sphericity (both from AMY [78]), and the differential 2-jet rate in the JADE scheme from AMY [79]. As before, our simulations in all four tunes agree with data within their uncertainties, but we observe some deviations in the mean values.

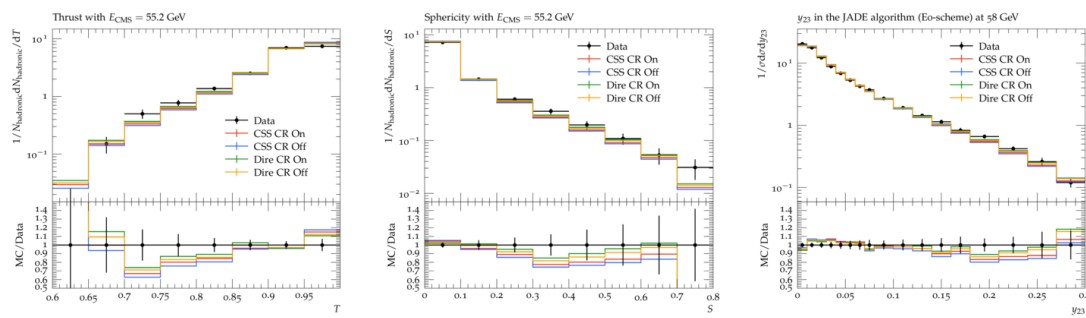

Figure 18: SHERPA results for thrust and sphericity compared with data from AMY [78] (left and centre), and for the differential 2-jet rate in the JADE scheme [79] (right).

We turn to particle spectra in Fig. 19, and display the energy-energy correlation (from

TOPAZ [80]), the longitudinal moment w.r.t. to the sphericity axis (from AMY [78]), and the scaled momentum spectrum of neutral kaons (from TOPAZ [81]). In all observables we notice the good agreement of the four SHERPA tunes with data.

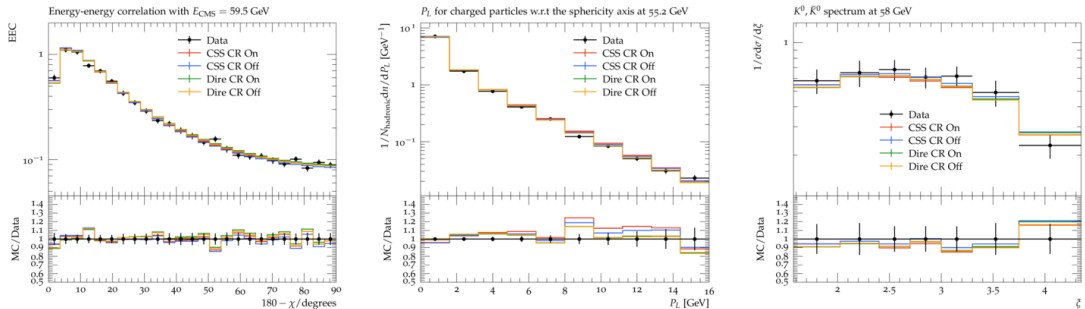

Figure 19: SHERPA results compared with data for the energy-energy correlation (left, from TOPAZ [80]), longitudinal moment w.r.t. the sphericity axis (centre, from AMY [78]) and the scaled momentum spectrum for neutral kaons (right, from TOPAZ [81]).

## 6 Summary

In this paper we have described, in detail, the re-implementation of the cluster fragmentation model within SHERPA, in an improved version compared to its original [27] publication. We have successfully tuned the model to data, for the two different parton showers available, and with or without the inclusion of a first, naive model for colour reconnections, which we also introduce here. We find, by and large, satisfying agreement of our model with data, with a slight preference for either using the DIRE [40] shower without colour reconnections or the CSShower [39] including them. This will facilitate future studies of further non–perturbative effects which may impact precision measurements of, e.g., the $W$ mass in hadronic final states at lepton colliders or of the top mass.

As a next step we will, in a future publication, investigate the impact of the new hadronisation model on those observables at hadron colliders that are susceptible to non–perturbative effects, including event and jet shape observables. This will also allow us to test the interplay of our model with the modelling of the underlying event.

## Acknowledgements

We would like to thank our colleagues from the SHERPA collaboration for the fruitful discussions and technical support. We would also like to thank S. Chhibra and H. Schulz for technical support. This work has received funding from the European Union's Horizon 2020 research and innovation programme as part of the Marie Skłodowska-Curie Innovative Training Network MCnetITN3 (grant agreement no. 722104). FK gratefully acknowledges funding as Royal Society Wolfson Research fellow.

## A  Hadron wavefunctions

The flavour parts of the charged meson wavefunctions are trivial, for example,

$$|\pi^+\rangle = |u\bar{d}\rangle\,, \tag{43}$$

while for the more complicated case of neutral mesons they are given by

$$
\begin{aligned}
|\pi^0\rangle, |\rho^0\rangle, \ldots, &= \frac{1}{\sqrt{2}}\left(|u\bar{u}\rangle - |d\bar{d}\rangle\right)\,, \\
|\eta^0\rangle, |\omega\rangle, \ldots, &= \frac{\cos\theta}{\sqrt{6}}\left(|u\bar{u}\rangle + |d\bar{d}\rangle - 2|s\bar{s}\rangle\right) - \frac{\sin\theta}{\sqrt{3}}\left(|u\bar{u}\rangle + |d\bar{d}\rangle + |s\bar{s}\rangle\right)\,, \\
|\eta'\rangle, |\phi\rangle, \ldots &= \frac{\sin\theta}{\sqrt{6}}\left(|u\bar{u}\rangle + |d\bar{d}\rangle - 2|s\bar{s}\rangle\right) + \frac{\cos\theta}{\sqrt{3}}\left(|u\bar{u}\rangle + |d\bar{d}\rangle + |s\bar{s}\rangle\right)\,, \\
|\eta_c\rangle, |J/\psi\rangle, \ldots &= |c\bar{c}\rangle\,, \\
|\eta_b\rangle, |\Upsilon(1s)\rangle, \ldots &= |b\bar{b}\rangle\,, \tag{44}
\end{aligned}
$$

and include the effect of singlet-octet mixing through suitable mixing angles, where SHERPA follows the recommendations of the PDG [82], *cf.* Tab. 1.

Table 1: Angles parameterising the singlet-octet mixing of mesons.

| Parameter | Description | Name (in run card) | Value |
|---|---|---|---|
| $\theta_{0^+}$ | mixing angle for the pseudoscalar multiplet | `Mixing_0+` | $-14.1°$ |
| $\theta_{1^-}$ | mixing angle for the vector multiplet | `Mixing_1-` | $36.4°$ |
| $\theta_{2^+}$ | mixing angle for the spin-2 multiplet | `Mixing_2+` | $27.0°$ |

The baryon wavefunctions are given by

$$
\begin{aligned}
|p\rangle &= \frac{1}{\sqrt{3}}|d(uu)_1\rangle + \frac{1}{\sqrt{6}}|u(ud)_1\rangle + \frac{1}{\sqrt{2}}|u(ud)_0\rangle\,, \\
|\Sigma^0_{(8)}\rangle &= \frac{1}{\sqrt{3}}|s(ud)_1\rangle + \frac{1}{\sqrt{12}}|d(su)_1\rangle + \frac{1}{\sqrt{4}}|d(su)_0\rangle + \frac{1}{\sqrt{12}}|u(sd)_1\rangle + \frac{1}{\sqrt{4}}|u(sd)_0\rangle\,, \\
|\Lambda_{(8)}\rangle &= \frac{1}{\sqrt{3}}|s(ud)_0\rangle + \frac{1}{\sqrt{12}}|d(su)_0\rangle + \frac{1}{\sqrt{4}}|d(su)_1\rangle + \frac{1}{\sqrt{12}}|u(sd)_0\rangle + \frac{1}{\sqrt{4}}|u(sd)_1\rangle\,, \\
|\Lambda_{(1)}\rangle &= \frac{1}{\sqrt{3}}|u(sd)_0\rangle + \frac{1}{\sqrt{3}}|d(su)_0\rangle + \frac{1}{\sqrt{3}}|s(ud)_0\rangle\,, \\
|\Delta^{++}\rangle &= |u(uu)_1\rangle\,, \\
|\Delta^+\rangle &= \sqrt{\frac{2}{3}}|d(ud)_1\rangle + \frac{1}{\sqrt{3}}|d(uu)_1\rangle\,, \\
|\Sigma^0_{(10)}\rangle &= \frac{1}{\sqrt{3}}|u(sd)_1\rangle + \frac{1}{\sqrt{3}}|d(su)_1\rangle + \frac{1}{\sqrt{3}}|s(ud)_1\rangle\,, \\
|\Lambda_Q\rangle &= |Q(qq)_0\rangle\,, \\
|\Sigma_Q\rangle &= |Q(qq)_1\rangle\,. \tag{45}
\end{aligned}
$$

A few comments are in order here. First, *all* decuplet hadrons, *i.e.* those that belong to a $\Delta$-like multiplet, are made up of spin-1 diquarks only. In addition, in some of the higher-lying multiplets, the usual octet is supplemented with a further singlet $\Lambda$ baryon, with the $\Lambda(1520)$ a good example. The wavefunction of these objects is totally symmetric and exclusively made of spin-0 diquarks. Finally, for baryons such as the neutron, the charged $\Sigma's$ or the $\Xi$'s of the octet multiplet, the wavefunctions emerge from the proton one by suitably replacing,

$$|n\rangle = \Big[|p\rangle\Big]_{u\leftrightarrow d} \ , \ |\Sigma^-\rangle = \Big[|p\rangle\Big]_{\substack{d\to s\\u\to d}} \ , \ |\Sigma^+\rangle = \Big[|p\rangle\Big]_{d\to s} \ , \ |\Xi^-\rangle = \Big[|p\rangle\Big]_{u\to s}$$

$$\text{and } |\Xi^0\rangle = \Big[|p\rangle\Big]_{\substack{u\to s\\d\to u}} . \tag{46}$$

## B Tuned Parameters

The tuning is performed with the PROFESSOR (v2.3.3) framework [45]. Approximately 10 million events are generated for each tune to ensure that the uncertainty in the SHERPA prediction in each bin is much smaller than the uncertainty in the data in the same bin.

### B.1 Parton shower parameters

Default parameters for the final state parton showers in SHERPA are listed in Tab. 2. With the exception of the infrared cut-off, which was tuned to data, all other parameters are fixed.

Table 2: (Fixed) perturbative input parameters for the parton showers and the tuned value for its infrared cut-off.

|  | Description | Name (in run card) | CSS | DIRE |
|---|---|---|---|---|
| $p_\perp^{(\text{cut})}$ | parton-shower cutoff | `CSS_FS_PT2MIN` | 1 | - |
| $\alpha_S(M_Z)$ | strong coupling in parton shower | `ALPHAS(MZ)` | 0.1188 | |
| | order for running $\alpha_S$ | `ORDER` | 2 | |
| $m_b^{(\text{pert})}$ | $b$ quark mass in parton shower | | 4.5 GeV | |
| $m_c^{(\text{pert})}$ | $c$ quark mass in parton shower | | 1.5 GeV | |
| $m_{u,d,s}^{(\text{pert})}$ | light quark masses in parton shower | | 0 GeV | |

### B.2 Constituent masses and popping parameters

In SHERPA, the quarks and diquarks have non-perturbative constituent masses that implicitly, through phase space, impact on their production in the forced decays of gluons at the end of the parton shower. While the quark masses are fixed directly, the diquark masses are calculated from the constituent masses of their component quarks as

$$m_{(ij)} = (m_i + m_j + m_{di}) \cdot (1 + \epsilon_{0,1}), \tag{47}$$

for spin-0 and spin-1 diquarks ($ij$) made of an $i$ and a $j$ quark. The (fixed) input parameters for all masses are listed in Tab. 3. When calculating the "popping" probabilities $\mathcal{P}$ for the

Table 3: (Fixed) non-perturbative input parameters for constituents: quark masses and parameters to calculate the diquark masses. Tuned popping probabilities for their non-perturbative production in gluon and cluster decays.

| | Description | Name (in run card) | Css CR Off (On) | DIRE CR Off (On) |
|---|---|---|---|---|
| $m_b$ | $b$ constituent mass | M_BOTTOM | 5.1 GeV | |
| $m_c$ | $c$ constituent mass | M_CHARM | 1.8 GeV | |
| $m_s$ | $s$ constituent mass | M_STRANGE | 0.4 GeV | |
| $m_{u,d}$ | $u$ & $d$ constituent masses | M_UP_DOWN | 0.3 GeV | |
| $m_g$ | gluon constituent masses | M_GLUE | 0.0 GeV | |
| $m_{di}$ | offset for diquark masses | M_DIQUARK_OFFSET | 0.3 GeV | |
| $\epsilon_0$ | rel. binding energy, spin-0 | M_BIND_0 | 0.12 | |
| $\epsilon_1$ | rel. binding energy, spin-1 | M_BIND_1 | 0.5 | |
| $p_s$ | strange quark probability | STRANGE_FRACTION | 0.46 | 0.4 |
| $p_{di}$ | diquark probability | BARYON_FRACTION | 0.15 (0.27) | 0.2 |
| $x_{qs}$ | $(qs)$ suppression | P_QS_by_P_QQ_norm | 0.71 | 0.71 |
| $x_{ss}$ | $(ss)$ suppression | P_SS_by_P_QQ_norm | 0.01 (0.013) | 0.02 |
| $x_1$ | $(qq)_1$ suppression | P_QQ1_by_P_QQ0_norm | 0.94 (0.63) | 0.57 |

constituents to be produced in gluon or cluster decays, SHERPA implicitly takes into account their number of spin states. As a consequence, up to a normalising their sum to unity, the individual $\mathcal{P}$ are given by

$$
\begin{aligned}
\mathcal{P}_{u,d} &= 2\,, & \mathcal{P}_s &= 2p_s\,, \\
\mathcal{P}_{(ud)_0} &= p_{di}\,, & \mathcal{P}_{(us)_0} = \mathcal{P}_{(ds)_0} &= x_{qs}p_s p_{di}\,, \\
\mathcal{P}_{(ud)_1} = \mathcal{P}_{(uu)_1} = \mathcal{P}_{(dd)_1} &= 3x_1 p_{di}\,, & \mathcal{P}_{(us)_1} = \mathcal{P}_{(ds)_1} &= 3x_{qs}p_s p_1 p_{di}\,, \\
\mathcal{P}_{(ss)_1} &= 3x_{ss}p_s^2 p_1 p_{di}\,.
\end{aligned}
\tag{48}
$$

The various parameters have been tuned to data and can be found in Tab. 3. It is worth noting that, at the moment, SHERPA does not feature any diquarks made of one or two heavy, *i.e.* charm or beauty, quarks.

## B.3 Kinematics

Table 4: Tuned parameters that describe kinematics in the non-perturbative decays of gluons and clusters, Eqs. (6), (7), and (22). * Note that in this publication we do not report on the tuning of the non-perturbative modelling for hadron colliders, which would involve additional physics modelling that we postpone to a future publication.

| | Description | Name (in run card) | Css CR Off (On) | Dire CR Off (On) |
|---|---|---|---|---|
| switch | select gluon splitting mode: 0: additive, 1: multiplicative, Eq. (7) | GLUON_DECAY_MODE | 0 | 0 |
| $\alpha_G$ | $\alpha$ in gluon decays, Eq. (7) | ALPHA_G | 0.67 | 0.67 |
| switch | select cluster splitting mode: 0: $z^{(i)}$, Eq. (22), 1: $\Delta M$ (exponential), 2: $\Delta M$ (Gaussian), 3: $\Delta M$ (log-normal), Eq. (26) | CLUSTER_SPLITTING_MODE | 0 | 0 |
| $\alpha_L$ | $\alpha$ (light quarks) in cluster fission | ALPHA_L | 2.5 | 2.5 |
| $\beta_L$ | $\beta$ (light quarks) in cluster fission | BETA_L | 0.13 | 0.12 |
| $\gamma_L$ | $\gamma$ (light quarks) in cluster fission | GAMMA_L | 0.27 (0.5) | 0.27 |
| $\alpha_D$ | $\alpha$ (diquarks) in cluster fission | ALPHA_D | 3.26 | 3.26 |
| $\beta_D$ | $\beta$ (diquarks) in cluster fission | BETA_D | 0.11 | 0.11 |
| $\gamma_D$ | $\gamma$ (diquarks) in cluster fission | GAMMA_D | 0.39 | 0.39 |
| $\alpha_H$ | $\alpha$ (heavy quarks) in cluster fission | ALPHA_H | 1.26 (3.55) | 1.26 |
| $\beta_H$ | $\beta$ (heavy quarks) in cluster fission | BETA_H | 0.98 (1.12) | 0.98 |
| $\gamma_H$ | $\gamma$ (heavy quarks) in cluster fission all in Eq. (22) | GAMMA_H | 0.05 (0.15) | 0.054 |
| $\alpha_B$ | $\alpha$ for decays of beam clusters* | ALPHA_B | 2.5 | 2.5 |
| $\beta_B$ | $\beta$ for decays of beam clusters* | BETA_B | 0.25 | 0.25 |
| $\gamma_B$ | $\gamma$ for decays of beam clusters* | GAMMA_B | 0.5 | 0.5 |
| $k_{\perp,0}$ | $k_{\perp,0}$ in gluon ($g \rightarrow f\bar{f}$) and cluster ($\mathcal{C} \rightarrow \mathcal{CC}$ and $\mathcal{C} \rightarrow \mathcal{HH}$) decays | KT_0 | 1.34 (1.42) | 1.34 |

Table 5: Tuned parameters used in the selection of the specific channel in $\mathcal{C} \to \mathcal{H}_1 + \mathcal{H}_2$ and $\mathcal{C} \to \mathcal{H}_1 + \gamma$ decays, *cf.* Eq. (31), including multiplet weights, and modifiers for individual hadrons or classes of hadrons.

| | Description | Name (in run card) | Css CR Off (On) | Dire CR Off (On) |
|---|---|---|---|---|
| switch | direct $\mathcal{C} \to \mathcal{H}$ enabled | DIRECT_TRANSITIONS | 1 | 1 |
| $x_\text{trans}$ | $\mathcal{C} \to \mathcal{H}$ threshold, Eq. (9) | TRANSITION_THRESHOLD | 0.51 (0.75) | 0.51 |
| $x_\text{dec}$ | $\mathcal{C} \to \mathcal{HH}$ threshold, Eq. (30) | DECAY_THRESHOLD | 0.02 (0.18) | 0.02 |
| $\chi$ | generic mass modifier, Eq. (31) | MASS_EXPONENT | 0 | 0 |
| meson multiplet weights (identified by the $\pi$-like meson) | | | | |
| $w_{000}$ | pseudoscalars ($\pi^\pm$, ...) | MULTI_WEIGHT_R0L0_PSEUDOSCALARS | 1 | 1 |
| $w_{001}$ | vectors ($\rho^{pm}$, ...) | MULTI_WEIGHT_R0L0_VECTORS | 2.5 | 2.2 |
| $w_{002}$ | tensors ($a_2(1320)^\pm$, ...) | MULTI_WEIGHT_R0L0_TENSORS2 | 1.5 | 1.5 |
| $w_{010}$ | scalars ($a_0(1450)^\pm$, ...) | MULTI_WEIGHT_R0L1_SCALARS | 0 | 0 |
| $w_{011}$ | axial vectors ($b_1(1235)^\pm$, ...) | MULTI_WEIGHT_R0L1_AXIALVECTORS | 0 | 0 |
| $w_{021}$ | axial vectors ($a_1(1260)^\pm$, ...) | MULTI_WEIGHT_R0L2_VECTORS | 0.5 | 0.5 |
| modifiers for specific mesons | | | | |
| $w_{M1}$ | singlet-meson modifier | SINGLET_MODIFIER | 2 | 2 |
| $w_\eta$ | $\eta$-meson modifier | ETA_MODIFIER | 2.2 (2.33) | 2.82 |
| $w_{\eta'}$ | $\eta'$-meson modifier | ETA_PRIME_MODIFIER | 4.5 (2.43) | 2.03 |
| baryon multiplet weights (identfied by some hadrons) | | | | |
| $w_{00\frac{1}{2}}$ | octet ($N(939)$, ...) | MULTI_WEIGHT_R0L0_N_1/2 | 1 | 1 |
| $w_{10\frac{1}{2}}$ | ($N(1535)$, ...) | MULTI_WEIGHT_R1L0_N_1/2 | 0.1 | 0.1 |
| $w_{20\frac{1}{2}}$ | ($N(1440)$, ...) | MULTI_WEIGHT_R1L0_N_1/2 | 0 | 0 |
| $w_{00\frac{3}{2}}$ | decuplet ($\Delta^{++}$, ...) | MULTI_WEIGHT_R0L0_DELTA_3/2 | 0.15 | 0.15 |
| modifiers for specific baryons | | | | |
| $w_{B1}$ | singlet-baryon modifier | SINGLETBARYON_MODIFIER | 1.8 | 1.8 |
| $w_{Bc}$ | $c$-baryon modifier | CHARMBARYON_ENHANCEMENT | 8 | 8 |
| $w_{Bb}$ | $b$-baryon modifier | BEAUTYBARYON_ENHANCEMENT | 0.8 | 0.8 |
| $w_{Bcs}$ | $cs$-baryon modifier | CHARMSTRANGEBARYON_ENHANCEMENT | 2 | 2 |
| $w_{Bbs}$ | $bs$-baryon modifier | BEAUTYSTRANGEBARYON_ENHANCEMENT | 1.4 | 1.4 |
| $w_{Bbc}$ | $bc$-baryon modifier | BEAUTYCHARMBARYON_ENHANCEMENT | 1 | 1 |

Table 6: Fixed mass thresholds for light cluster to hadron decays.

| | Description | Name (in run card) | Value |
|---|---|---|---|
| $M_{\pi\gamma}$ | mass threshold for $\mathcal{C} \to \pi^0 + \gamma$ | PI_PHOTON_THRESHOLD | 0.150 GeV |
| $M_{\pi\pi}$ | mass threshold for $\mathcal{C} \to \pi + \pi$ | DI_PION_THRESHOLD | 0.300 GeV |

### B.4 Colour Reconnections

In Tab. 7 we list some of the parameters that describe the simple colour reconnection model in SHERPA. It should be noted though that

1. we added the model as an additional option after the parameters for the overall hadronisation model had been fitted to LEP data, so a better tune may be achieved by a combined re-fitting exercise;

2. the impact of colour reconnections on hadron-level observables in $e^-e^+$ annihilations is moderate: the parton shower usually terminates with only few partons in the final state which are relatively tightly correlated in colour and momentum space already; and that

3. the spatial component of the model becomes accessible in hadron collision only.

Table 7: Tuned parameters in the simple colour reconnection model provided in SHERPA, Section 3. * Note that the spatial part of the model is relevant for hadron collisions only and will be tuned in conjunction with a forthcoming tuning of SHERPA's model for the underlying event.

| | Description | Name (run card) | CSS | DIRE |
|---|---|---|---|---|
| switch | switching colour reconnections on/off | MODE | On/Off | |
| switch | select colour reconnection mode: 0: logarithmic 1: power, *cf.* Eq. (39) | PMODE | 0 | |
| $Q_0$ | momentum space distance, Eq. (39) | Q_0 | 1.41 GeV | 1.65 GeV |
| $\eta_Q$ | scaling parameter for $Q_0$, Eq. (39) | etaQ | 0.16 | |
| $R_0$ | transverse space distance, Eq. (40) * | R_0 | 1.0/GeV | |
| $\eta_R$ | scaling parameter for $R_0$, Eq. (39) * | etaR | 0.16 | |
| $\kappa_C$ | colour weight for different colours | Reshuffle | 0.33 | |
| $\kappa$ | exponent in the norm, Eq. (42) | kappa | 2 | |

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
