# Peer review of "Cluster Hadronisation in Sherpa"

_SciPost Physics, doi:SciPost Phys. 13, 019 (2022)_

## Round 1 · Referee Report · Anonymous (Referee 1) · 2022-4-26

Strengths

1 - Concise summary of the cluster hadronization model as implemented/reimplemented in SHERPA.
2 - Comprehensive comparison to electron-positron data.

Weaknesses

1 - The notation is sometimes a little too unwieldy.
2 - Citations of similar advances in the field lacking.
3 - A large number of small typos (though while time-consuming, are simple to fix)

Report

The paper is a concise summary of the cluster model as implemented in the Monte Carlo event generator Sherpa. It is an update of the previous version of the cluster model Sherpa used, and all of the phenomenological components of the model are presented. Beginning with how a cluster is formed at the end of the shower, the authors then discuss the cluster fission and decay steps whereby clusters are converted into (unstable) hadrons. The authors then present their implementation of a colour reconnection algorithm, and finally present a selection of tuned parameter runs for $e^+e^-$ collisions.

Hadronization is often an overlooked part of Monte Carlo event generation and the cluster model presented is different enough from the one implemented in Sherpa that the work is justified in being published. I would happily recommend it for publication after the authors address the points I have written below.

Requested changes

Citations I think are missing: - Papers by Bierlich and collaborators about colour reconnection and rope hadronization. - Fischer & Sjöstrand's paper on thermodynamic string breaking and Chakraborty et. al's paper on rope shoving. - Papers by the Herwig collaboration that have added features to the cluster model e.g. Kirchgäßer et. al's paper on Baryonic colour reconnection or their paper 'Soft and Diffractive Scattering in Cluster Hadronization', and others. - Eq. (2.22) looks very similar to the string fragmentation formula for selecting the longitudinal momentum fraction. I would suggest a reference at this point. - Section 3 on colour reconnection has close to no references. There are a variety of papers (some of which I have mentioned above) that deserve a mention. Others include Bellm's paper on ColouRea, Lönnblad et. al's papers on dipole swing, Skands et. al's papers on colour reconnection, Sjöstrand and Khoze's paper on investigating colour reconnection in $W^+W^-$ events.

Typos and/or formatting: General: 1 - Hadronization is spelt with a 'z' everywhere in the paper apart from the title. 2 - I think the use of a hyphen in 'di-quark' and 'pop-corn' is unconventional. Diquarks are familiar enough that authors have (in general) dropped the hyphen. You also miss out the hyphen later in the paper.

Section-specific: 3 - Page 2, last paragraph of Section 2.1.2, the quotation mark for "splitter" needs to be fixed. 4 - The normalization of the decomposition in Eq. (2.3) would, it seems to me, imply that $p_1^2 = \frac{m_{f_1}^2}{2}$. I think you need to have the prefactor of both vectors be $\frac{M_{12}}{\sqrt{2}}$, rather than $\frac{M_{12}}{2}$ 5 - Typo for $\mu$ in positive light-like vector $n^{\mu}_+$ for $p_2$ in Eq. (2.3) 6 - I would like some linking of the variables appearing e.g. $\alpha$ in Eq. (2.7) with the table of parameters in the appendix, or at least some consistency in how they are labeled. Is the $\alpha_G$ in Tab. 4 this $\alpha$ or is it another? 7 - Missing bracket for $z^{(0)}$ in last paragraph of Section 2.1.4.

8 - First sentence of Section 2.1.7, I think you wish to use 'transition' instead of 'transit'. 9 - Missing slash in last line of the first case of anomalies: $p_1\to p'_1 ...$ 10 - In the third case of the anomalies, I don't think you mean to use 'can directly $\textit{transfer}$ '[emphasis my own]. I believe you mean 'transition'. 11 - I would like Figure 1 to be enlarged to the width of the text (or close to). 12 - In the first paragraph of Section 2.1.8, there is no reference to what the tuned values are for the parameters. Perhaps direct the reader to the appendix.

13 - In Section 2.2.1, I find the sentence: "in SHERPA, diquarks are allowed to constitute clusters" to be cumbersome and poorly worded. Do you mean that they are allowed to be constutents of clusters? If so, I would also like a reference to the above mentioned paper by Gieseke, Kirchgaesser, and Plaetzer on Baryonic Colour Reconnection. 14 - In the decomposition of the momenta in Section 2.2.2, I would check the normalization as before. 15 - In the paragraph between Equations (2.21) and (2.22), 'Sherpa offers $\textit{both}$ methods' should probably be 'offers two methods'. 16 - Last line of Section 2.2.2 should read 'the $\textit{azimuthal }$ angle'. Also the typesetting of the last part of the second method of cluster fragmentation shouldn't have such wide margins. 17 - Title of Section 2.2.3 should be 'transitioning into', not 'transiting to'. In the same section, the sentence "decays into one or two hadrons plus one or no clusters" is not only quite awkward phrasing, but it also implies to me the possible situation of 2 hadrons and a cluster could be produced, which I doubt is what you meant. I would simply write it more explicitly: e.g. 'into two hadrons, one hadron and a cluster, or one hadron.'

18 - In Section 2.2.4, missing slash in first definition of $x_p$ in paragraph below Fig.2. Also, you have defined $\vec{p}_f$, but not $\vec{p}_C$. I'm guessing it's the cluster's momentum, but I would like to see that written down. 19 - In Section 2.2.4, leading quotation marks for "fragmentation function" and "fragmentation" are incorrect. In the same section, the sentence "We observe,..., as simple result ..." There are several words too many or too few here. Also, the next sentence "This manifest$\textit{s }$ itself" [emphasis mine] 20 - Section 3 on colour reconnection has 0 justification as to why it is a necessary component of hadronization, particularly for hadron-hadron collisions. A small bit (i.e. one paragraph) about what colour reconnection is would be necessary. As it stands it seems like it offers the mere possibility to swap colour connections between partons without any physical intuition behind it. Why do the authors choose to use a 'distance' measure instead of cluster mass? Some kind of justification/clarification is needed. Also the quotation marks for "swapping probability" are incorrect. 21 - A point that should be made by the authors is when the colour reconnection algorithm/step takes place. It seems like it occurs before cluster creation, which is an important point to make because the Herwig colour reconnection algorithm occurs after the cluster have been created. 22 - Section 4, first sentence: 'electron-position' should be "positron".

23 - Section 4, "By far and large" should be "By and large", and in the same sentence "satisfying" should be "satisfactory". 24 - Given that this work has focused on the hadronization models, yet there is a large amount of comparison between two different showers (CSShower and Dire), I would recommend a small section outlining the major differences between the two. 25 - Section 4.1, same remark with the "By far and large". 26 - Section 4.1, typesetting of $N_{ch} > 40$. The inequalities need to be lowered to be better in-line.- There is no mention of the cluster fission and decay work that Kupco 27 - Section 4.1, last sentence, "undrshoot" should be "undershoot".

28 - Section 4.2, comma missing after "(or more precisely $1-T$)$\textbf{,}$" 29 - Section 4.4, same point about "By far and large" 30 - Section 4.4, same point about the in-line inequality. 31 - Section 4.4, last paragraph on page 15, "shoes" should read "shows". 32 - Section 4.5, the discussion about the production of diquarks at the cluster fission and decay level is fair to say, but does not acknowledge the possibility of producing baryonic clusters via colour reconnection vis-a-vis Geiseke, Kirchgaesser, and Plaetzer.

33 - Section 5, last sentence "electon-positron" should read "electron-positron". 34 - Section 6, same point as above about "by far and large". 35 - A point that I have mentioned above implicitly, but would like to make explicit here is that there is virtually no mention of the authors' appendices anywhere in the main body of the text. The clarity of the paper suffers as a result. I would like to see a better linking of the two parts. 36 - In Equation (B.1), the $x_{0,1}$ are, I suppose, the relative binding energies for spin-0 and spin-1 diquarks from Table 3 respectively. Do they really have dimensions of mass? Equation (B.1) would suggest they are dimensionless. Also, there is a degeneracy in the authors' notation as five rows later in Table 3, they use $x_1$ again, to denote a completely different quantity. I would suggest choosing a different variable, simply for the sake of clarity.
37 - Section 2.3.2, in case 2, the sentence "Clusters, that are too..., will decay ..." should read "Clusters too light ... will decay", i.e. remove the commas and the "that are".

  • validity: top
  • significance: good
  • originality: good
  • clarity: good
  • formatting: good
  • grammar: good

Author:  Gurpreet Singh Chahal  on 2022-06-16  [id 2584]

(in reply to Report 1 on 2022-04-26)
Category:
answer to question
correction

Thank you very much for the detailed suggestions. All the comments have been addressed and a new version v2 has been submitted to arXiv, which includes all the updates.

Weaknesses

1 . The notation is sometimes a little too unwieldy. - Additional explanation added wherever needed

2 . Citations of similar advances in the field lacking. - All the relevant suggested references are added

3 . A large number of small typos (though while time-consuming, are simple to fix) - All typos are fixed

Requested changes Typos and/or formatting:

  • corrected the equation 2.3
  • 6: updated tables
  • 13: added reference to Gieseke, Kirchgaesser and Plaetzer as a footnote, and modified the sentence, see above.
  • 14: equation 2.18 and 2.19 are updated
  • 18-19: fixed and added a description of p_c
  • 20-21: fixed and added a sentence in Section 3 to the first paragraph.
  • 24: CSShower vs. Dire - we believe this is outside the scope of the paper.
  • 36: renamed $x_{0.1}$ as $\epsilon_{0,1}$ in table 3 and Eq. (B.1)

---

## Round 1 · Referee Report · Anonymous (Referee 2) · 2022-6-1

Strengths

1- good description of model 2- thorough discussion of comparison to experimental data

Weaknesses

1- not too much references to other recent work on the subject 2- only little details on tuning 3 - little discussion of relative importance of parameters

Report

The authors present a new implementation of the cluster hadronization
model in the Monte Carlo event generator Sherpa. The model finds colour
singlet clusters of partons after a splitting of gluons into
flavour-anti flavour pairs. These clusters commonly decay into hadron
pairs according to weights that reflect the available phase space,
flavour multiplet and spin weights as well as some special weights that
can be encoded in the wave-function for individual hadrons. A detailed
discussionn of how exceptional cases at the lower threshold for hadron
pair production are being treated is given where the model may fall back
to a decay of clusters to individual hadrons and perhaps additional
photons in order to capture unbalanced four momenta. The model has been
tuned to data from hadron production at electron positron colliders,
first and foremost at the Z pole and a consitency check for various lower
energies is presented.

The paper is written clearly, all details are presented thoroughly and
the comparison with experimental data is discussed in great detail. The
paper should be published in SciPost, however, in a few places some more
details and discussion could be given to round off the presentation.
They are listed below, together with a few minor corrections.

Requested changes

  • the figures are quite small and some annotations difficult to read on a paper printout.

  • p1, second paragaph: "exact" and "fixed order" are somewhat contradictory. Perhaps "exact" should be replaced with "perturbative".

  • While the introduction cites older hadronazation models, a lot of more recent work on hadronization models, also in the context of colour dynamics has been left unnoticed. References to the work of the Lund and Herwig groups on e.g. colour rope formation and colour reconnection would complete a contemporary review on hadronization models that belongs to the introduction of such an article.

  • Eq.(2.3) m -> mu

  • p3, bottom: "splits one of the two gluons" probably means taking the other gluon as a spectator as previously a quark has been.

  • p4, 2): typo "to"

  • sect 2.2.1: diquarks are still produced strongly correlated, aren't they? It is only their larger momentum as they have been produced in primary clusters, that will somewhat smear out baryon pairs.

  • p6, bottom: "shower" should probably be removed.

  • p8, 2nd line under figure: "vec"

  • p8, last paragraph: "decreases with the mass" sounds like decreasing with decreasing mass, but the opposite is meant.

  • 2.3.1: "triplet flavour" may easily be confused with "flavour triplet" in this context. Perhaps "colour triplet" should be written explicitly.

  • Eq.(2.31): what is "\cal H"? Should be some mass but is left without explanation.

  • 2.3.3.: The longitudinal decay kinematics have some impact on later results? x_p?

  • Eq.(3.3) x_perp not explained and unclear how they would be assigned to partons in the first place.

  • Sect 4: the presentation of results should probably be preceded by a few comments on the tuning strategy and methodology. Many parameters have been introduced. Are they all equally important?

  • Sect 4.2, end: the use of colour reconnection is commented on briefly. They should only have limited impact in e+e- annihilation.

  • 4.5, end: "cluster" -> "clusters"

  • p20, top: "satisfactory" -> "satisfactorily"

  • validity: high
  • significance: top
  • originality: high
  • clarity: high
  • formatting: good
  • grammar: good

Author:  Gurpreet Singh Chahal  on 2022-06-17  [id 2588]

(in reply to Report 2 on 2022-06-01)

Thank you so much for the detailed comments. All the suggestions have been addressed and the updated draft resubmitted as v2 (https://arxiv.org/pdf/2203.11385v2.pdf)

Weaknesses 1. not too much references to other recent work on the subject - extended introduction by adding about 20 more references

2 only little details on tuning - Added more details

3 little discussion of relative importance of parameters - importance of parameters explained

Requested changes

  1. All typos are fixed

  2. the figures are quite small and some annotations difficult to read on a paper printout.

    • The size of the figures has been selected appropriately to show a large number of figures, and people can zoom in if needed
  3. While the introduction cites older hadronazation models, a lot of more recent work on hadronization models, also in the context of colour dynamics has been left unnoticed. References to the work of the Lund and Herwig groups on e.g. colour rope formation and colour reconnection would complete a contemporary review on hadronization models that belongs to the introduction of such an article.

    • References added
  4. p3, bottom: "splits one of the two gluons" probably means taking the other gluon as a spectator as previously a quark has been.

    • Fixed: p3, bottom:added ", with the other gluon acting as spectator"
  5. sect 2.2.1: diquarks are still produced strongly correlated, aren't they? It is only their larger momentum as they have been produced in primary clusters, that will somewhat smear out baryon pairs.

    • added a sentence for clarification: "Allowing diquark production at every stage in the hadronisation process, \ie in both gluon decays and in the subsequent fission of clusters into secondary clusters, softens their strong correlation." and reformulated " represents a dynamic realisation " as "This represents an alternative the popcorn mechanism~\cite{Andersson:1984af,Eden:1996xi} in the Lund string model within cluster Hadronisation models, which softens the previous strong correlation of baryon--anti-baryon pairs."
  6. p8, last paragraph: "decreases with the mass" sounds like decreasing with decreasing mass, but the opposite is meant.

    • sentence updated as “decreases with increasing mass”
  7. 2.3.1: "triplet flavour" may easily be confused with "flavour triplet" in this context. Perhaps "colour triplet" should be written explicitly.

    • updated the sentence accordingly
  8. Eq.(2.31): what is "\cal H"? Should be some mass but is left without explanation.

    • Fixed: Eq (2.31) - yes, corrected "\cal{H}i" to "m}_i
  9. 2.3.3.: The longitudinal decay kinematics have some impact on later results? x_p?

    • yes, they do, for example when you think about heavy quark fragmentation functions.
  10. Eq.(3.3) x_perp not explained and unclear how they would be assigned to partons in the first place.

    • added for clarification: "Note that spatial distances are relevant only in cases like, for example, hadron---hadron collisions, where the individual scatters of the underlying event can occur at different positions in the transverse plane."
  11. Sect 4: the presentation of results should probably be preceded by a few comments on the tuning strategy and methodology. Many parameters have been introduced. Are they all equally important?

    • Fixed: added a half-paragraph on the relevant explanation
  12. Sect 4.2, end: the use of colour reconnection is commented on briefly. They should only have limited impact in e+e- annihilation.

    • The impact is limited, modified the last sentence accordingly by adding the word "slightly".

---

## Round 2 · Referee Report · Anonymous (Referee 2) · 2022-6-18

Report

All comments from the first report have been addressed.

---

## Round 2 · Referee Report · Anonymous (Referee 1) · 2022-6-20

Report

I am happy to recommend the updated manuscript for publication. The authors have addressed all the points in my previous report.

---

## Round 2 · Author Response

All the comments received on version 1 have been addressed in this version 2 of the paper.

---

## Round 2 · List of Changes

The detailed list of changes have been mentioned as replies to the referee comments.

---

## Editorial Decision

published